# Multi-Step Generalized Policy Improvement by Leveraging Approximate Models

**Lucas N. Alegre**[1,2]    **Ana L. C. Bazzan**[1]    **Ann Nowé**[2]    **Bruno C. da Silva**[3]

[1]Institute of Informatics, Federal University of Rio Grande do Sul

[2]Artificial Intelligence Lab, Vrije Universiteit Brussel    [3]University of Massachusetts

{lnalegre,bazzan}@inf.ufrgs.br   ann.nowe@vub.be   bsilva@cs.umass.edu

## Abstract

We introduce a principled method for performing zero-shot transfer in reinforcement learning (RL) by exploiting approximate models of the environment. Zero-shot transfer in RL has been investigated by leveraging methods rooted in generalized policy improvement (GPI) and successor features (SFs). Although computationally efficient, these methods are model-free: they analyze a library of policies—each solving a particular task—and identify which action the agent should take. We investigate the more general setting where, in addition to a library of policies, the agent has access to an *approximate* environment model. Even though model-based RL algorithms can identify near-optimal policies, they are typically computationally intensive. We introduce $h$-GPI, a multi-step extension of GPI that interpolates between these extremes—standard model-free GPI and fully model-based planning—as a function of a parameter, $h$, regulating the amount of time the agent has to reason. We prove that $h$-GPI's performance lower bound is strictly better than GPI's, and show that $h$-GPI generally outperforms GPI as $h$ increases. Furthermore, we prove that as $h$ increases, $h$-GPI's performance becomes arbitrarily less susceptible to sub-optimality in the agent's policy library. Finally, we introduce novel bounds characterizing the gains achievable by $h$-GPI as a function of approximation errors in both the agent's policy library and its (possibly learned) model. These bounds strictly generalize those known in the literature. We evaluate $h$-GPI on challenging tabular and continuous-state problems under value function approximation and show that it consistently outperforms GPI and state-of-the-art competing methods under various levels of approximation errors.

## 1   Introduction

Reinforcement learning (RL) (Sutton and Barto, 2018) algorithms have achieved remarkable performance in challenging tasks both in the model-free (Bellemare et al., 2020; Wurman et al., 2022) and the model-based (Bryant et al., 2022; Wu et al., 2022) settings. In these problems, agents are typically trained to optimize one particular reward function (a *task*). However, designing agents that can adapt their decision-making policies to solve novel tasks in a zero-shot manner—i.e., without requiring any further learning—remains an important open problem.

Various principled and efficient policy transfer methods, which combine previously-acquired policies to solve new tasks in a zero-shot manner, have been proposed based on the combination of successor features (SFs) and generalized policy improvement (GPI) (Barreto et al., 2017; Borsa et al., 2019; Barreto et al., 2020; Kim et al., 2022). On the one hand, SFs allow agents to efficiently evaluate the performance of given policies in arbitrary sets of tasks. On the other hand, GPI extends the classic policy improvement procedure (Sutton and Barto, 2018) by analyzing a library of previously-learned

37th Conference on Neural Information Processing Systems (NeurIPS 2023).

policies—each solving a particular task—and identifying a novel policy that simultaneously improves upon all policies in the agent's library.

We introduce $h$-GPI, a multi-step extension of GPI that interpolates between these extremes——standard model-free GPI and fully model-based planning——as a function of a parameter, $h$, regulating the amount of time the agent has to reason. An $h$-GPI policy is computed by performing online planning for $h$ steps, using an approximate model, and then employing GPI on all states reachable in $h$ steps to identify (in a zero-shot manner) which actions the agent should take thereafter. See Figure 1 for an example.

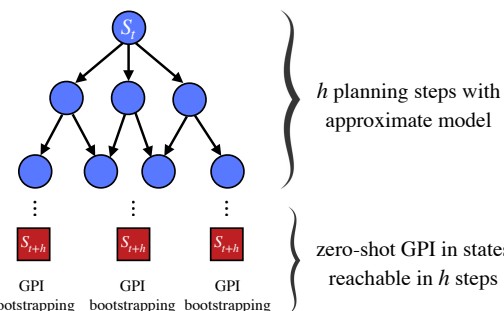

Figure 1: $h$-GPI combines online planning with GPI leveraging a learned model.

Our work is partially motivated by the observation that state-of-the-art model-based RL algorithms (Hafner et al., 2019, 2021) often adapt poorly to local changes in the environment (Wan et al., 2022). Furthermore, environment models may have been learned and thus might be inaccurate. These observations underscore the risks involved in relying solely on approximate models for long-horizon planning scenarios, as model errors compound and may lead to catastrophic outcomes. Unlike techniques for dealing with approximate models, our method additionally exploits GPI's capabilities of efficiently performing zero-shot transfer in a model-free manner. Other approaches—similar in nature to $h$-GPI—make use of models and GPI-based techniques. Thakoor et al. (2022), for example, introduced GGPI, a method that learns state visitation models induced by particular policies to rapidly evaluate policies under a given, known reward function. The authors show that performing GPI over a particular type of non-stationary policy produces behaviors that outperform those in the agent's library policy. $h$-GPI, by contrast, learns a different type of model: an environment model, which is used to perform planning—i.e., action selection—rather than policy evaluation. Moreover, $h$-GPI exploits GPI to perform bootstrapping from all states reachable in $h$ steps. In Section 5, we contrast our method with other related techniques, highlighting the unique aspects of our approach.

In this paper, we introduce the first principled method capable of exploiting approximate models and producing a multi-step extension of GPI with formal guarantees. We prove that $h$-GPI's performance lower bound is strictly better than GPI's, and show that $h$-GPI generally outperforms GPI as $h$ increases. Furthermore, we prove that as $h$ increases, $h$-GPI's performance becomes arbitrarily less susceptible to value function approximation errors in the agent's policy library. Finally, we introduce novel bounds characterizing the gains achievable by $h$-GPI as a function of approximation errors in both the agent's policy library and its (possibly learned) model. We empirically evaluate $h$-GPI in challenging tabular and continuous-state problems with value function approximation. Our findings show that it consistently outperforms both GPI and other state-of-the-art methods under various levels of approximation errors. These results, combined with our method's formal guarantees, indicate that $h$-GPI is an important first step towards bridging the gap between model-free GPI-based methods and fully model-based planning algorithms—while being robust to approximate models of the environment and value function estimation errors.

## 2 Background

Before introducing our contributions, we review key concepts and definitions related to model-free and model-based RL, SFs, and GPI.

### 2.1 Reinforcement learning

RL problems (Sutton and Barto, 2018) are typically modeled as *Markov decision processes* (MDPs). An MDP is a tuple $M \triangleq (\mathcal{S}, \mathcal{A}, p, r, \mu, \gamma)$, where $\mathcal{S}$ is a state space, $\mathcal{A}$ is an action space, $p(\cdot|s,a)$ denotes the distribution over next states conditioned on a state and action, $r : \mathcal{S} \times \mathcal{A} \times \mathcal{S} \mapsto \mathbb{R}$ is a reward function, $\mu$ is an initial state distribution, and $\gamma \in [0, 1)$ is a discount factor. Let $S_t$, $A_t$, and $R_t = r(S_t, A_t, S_{t+1})$ be random variables corresponding to the state, action, and reward, respectively,

at time step $t \geq 0$. The goal of an RL agent is to learn a policy $\pi : \mathcal{S} \mapsto \mathcal{A}$ that maximizes the expected discounted sum of rewards (*return*) $\sum_{t=0}^{\infty} \gamma^t R_t$. The action-value function of a policy $\pi$ is $q^\pi(s, a) \triangleq \mathbb{E}_\pi[\sum_{i=0}^{\infty} \gamma^i R_{t+i} | S_t = s, A_t = a]$, where $\mathbb{E}_\pi[\cdot]$ denotes the expectation over trajectories induced by $\pi$ and $p$. Given $q^\pi$, a *greedy* policy can be defined as $\pi'(s) \in \arg\max_a q^\pi(s, a)$. It is guaranteed that $q^{\pi'}(s, a) \geq q^\pi(s, a)$, for all $(s, a) \in \mathcal{S} \times \mathcal{A}$. Computing $q^\pi$ and $\pi'$ is done by processes known, respectively, as *policy evaluation* and *policy improvement*. Repeatedly alternating between policy evaluation and policy improvement steps is known to lead to an optimal policy, $\pi^*(s)$, which maximizes the expected return from all states $s \in \mathcal{S}$ (Puterman, 2014).

## 2.2 Successor features and GPI

The *successor features* (SFs) framework (Barreto et al., 2017) allows agents to efficiently evaluate the performance of a given policy when deployed under any linearly-representable reward functions, $r_\mathbf{w}(s, a, s') = \boldsymbol{\phi}(s, a, s') \cdot \mathbf{w}$, where $\boldsymbol{\phi}(s, a, s') \in \mathbb{R}^d$ are reward features and $\mathbf{w} \in \mathbb{R}^d$ are weights. A controlled Markov process $(\mathcal{S}, \mathcal{A}, p, \mu, \gamma)$, i.e., an MDP without a reward function, when combined with reward features $\boldsymbol{\phi} : \mathcal{S} \times \mathcal{A} \times \mathcal{S} \mapsto \mathbb{R}^d$, induces a family of MDPs:

$$\mathcal{M}^\phi \triangleq \{M = (\mathcal{S}, \mathcal{A}, p, r_\mathbf{w}, \mu, \gamma) \mid r_\mathbf{w}(s, a, s') = \boldsymbol{\phi}(s, a, s') \cdot \mathbf{w}\}. \tag{1}$$

Notice that the family of MDPs $\mathcal{M}^\phi$ represents all possible tasks (each defined by a reward function) that can be defined in the environment associated with the corresponding controlled Markov process. Given a policy $\pi$, its corresponding SFs, $\boldsymbol{\psi}^\pi(s, a) \in \mathbb{R}^d$, for a state-action pair $(s, a)$ are defined as

$$\boldsymbol{\psi}^\pi(s, a) \triangleq \mathbb{E}_\pi \left[ \sum_{i=0}^{\infty} \gamma^i \boldsymbol{\phi}_{t+i} \mid S_t = s, A_t = a \right], \tag{2}$$

where $\boldsymbol{\phi}_t \triangleq \boldsymbol{\phi}(S_t, A_t, S_{t+1})$. Importantly, notice that given the SFs $\boldsymbol{\psi}^\pi(s, a)$ of a policy $\pi$, it is possible to *directly* compute the action-value function $q_\mathbf{w}^\pi(s, a)$ of $\pi$, under *any* linearly-expressible reward functions, $r_\mathbf{w}$, as follows:[1]

$$q_\mathbf{w}^\pi(s, a) = \mathbb{E}_\pi \left[ \sum_{i=0}^{\infty} \gamma^i \boldsymbol{\phi}_{t+i} \cdot \mathbf{w} \mid S_t = s, A_t = a \right] = \boldsymbol{\psi}^\pi(s, a) \cdot \mathbf{w}. \tag{3}$$

The equation described above—which uses SFs to evaluate a given policy under different reward functions—represents a process known as *generalized policy evaluation* (GPE) (Barreto et al., 2020). Notice that this process can be extended to the case where an agent has access to a set of previously-learned policies, $\Pi = \{\pi_i\}_{i=1}^n$, and their corresponding SFs, $\Psi = \{\boldsymbol{\psi}^{\pi_i}\}_{i=1}^n$. Then, given any $\pi_i \in \Pi$, GPE can be used to efficiently evaluate $\pi_i$ under any arbitrary reward function of interest, $r_\mathbf{w}$, via Equation (3): $q_\mathbf{w}^{\pi_i}(s, a) = \boldsymbol{\psi}^{\pi_i}(s, a) \cdot \mathbf{w}$.

**Generalized policy improvement.** GPI generalizes the standard policy improvement process (discussed in Section 2.1). It assumes access to a *set* of policies (and corresponding action-value functions) and uses it to directly identify a novel policy optimizing a particular reward function, $r_\mathbf{w}$. Importantly, the novel policy is guaranteed to outperform all original policies the agent had access to.

**Definition 1.** *(Barreto et al., 2020) Given a set of policies $\Pi$ and a reward function $r_\mathbf{w}$, generalized policy improvement (GPI) is the process by which a policy, $\pi'$, is identified such that*

$$q_\mathbf{w}^{\pi'}(s, a) \geq \max_{\pi \in \Pi} q_\mathbf{w}^\pi(s, a) \text{ for all } (s, a) \in \mathcal{S} \times \mathcal{A}. \tag{4}$$

Based on Equation (4) and the reward decomposition $r_\mathbf{w}(s, a, s') = \boldsymbol{\phi}(s, a, s') \cdot \mathbf{w}$, a *generalized policy*, $\pi : \mathcal{S} \times \mathcal{W} \mapsto \mathcal{A}$, can then be defined as follows:

$$\pi^{\text{GPI}}(s; \mathbf{w}) \in \arg\max_{a \in \mathcal{A}} \max_{\pi \in \Pi} q_\mathbf{w}^\pi(s, a). \tag{5}$$

Let $q_\mathbf{w}^{\text{GPI}}(s, a)$ be the action-value function of policy $\pi^{\text{GPI}}(\cdot; \mathbf{w})$. The GPI theorem (Barreto et al., 2017) shows that $\pi^{\text{GPI}}(\cdot; \mathbf{w})$ satisfies Definition 1, and so Equation (5) can be used to identify a policy guaranteed to perform at least as well as any other policies $\pi_i \in \Pi$, when tackling a new task, $\mathbf{w}$. This theorem can be extended to cases where the agent only has access to an approximation of the action-value function, $\hat{q}^{\pi_i}$, of a policy $\pi_i$ (Barreto et al., 2018).

---

[1]Notice that the definition of SFs satisfies a form of the Bellman equation, where the features $\boldsymbol{\phi}_t$ play the role of rewards. Thus, SFs can be learned through standard temporal-difference learning algorithms.

## 2.3 Model-based RL

In model-free RL, agents do not have access to the environment state transition and reward functions, $p$ and $r$, and must, instead, learn a policy solely based on samples obtained while interacting with the environment. Given an MDP, we denote $m \triangleq (p, r)$ as its *model*. In *model-based* RL, an agent is given (or has to learn) an approximate model, $\hat{m} \approx m$. Agents may exploit models in different ways; for example, *(i)* to perform *background* planning (i.e., generating simulated experiences to more rapidly learn a policy or value function, as in Dyna-style algorithms (Van Seijen and Sutton, 2013; Janner et al., 2019; Pan et al., 2019)); and *(ii)* to perform *online* planning (Barto et al., 1995; Chua et al., 2018; Efroni et al., 2020; Hansen et al., 2022).

In online planning methods, agents use a model to estimate the value of states by simulating the outcomes of a particular sequence of actions. This is also known as "unrolling the model forwards", from the current state, for a given number of time steps. Typically, online planning procedures are performed from every state the agent encounters, as in model-predictive control (MPC) algorithms (Camacho and Alba, 2013; Chua et al., 2018). Importantly, recent work—such as that of Wan et al. (2022)—has shown that state-of-the-art model-based RL algorithms (Hafner et al., 2019, 2021) often fail catastrophically when deployed in settings where the environment (e.g., the MDP's reward function) may change. This observation underscores the risks involved in relying solely on approximate models for long-horizon planning scenarios, since it is well-known that model errors compound.

We address these limitations by introducing a zero-shot transfer technique based on online planning. Our technique interpolates between standard model-free GPI and fully model-based planning according to a parameter, $h$, regulating the amount of time the agent has to reason using an approximate model. Unlike previous methods that require re-learning or adapting models to tackle novel tasks, our technique exploits approximate models and SFs to solve any tasks (defined as linearly-expressible reward functions) *without requiring any further learning*. Importantly, it has strong theoretical guarantees regarding its performance lower bound under various types of approximation errors.

## 3 Multi-step generalized policy improvement

We now introduce our main contribution, $h$-GPI. This is a multi-step extension of GPI that combines online planning with approximate environment models to efficiently perform zero-shot transfer.

We start by defining the Bellman optimality operator $\mathcal{T}^*$ (and its multi-step variant) applied to action-value functions $q \in \mathcal{Q}$, where $\mathcal{Q}$ is the space of action-value functions. To simplify notation, given a model $m \triangleq (p, r)$, we denote $\mathbb{E}_m[\cdot]$ as the expectation operator with respect to $S_{t+1} \sim p(.|S_t, A_t)$ and $R_t \triangleq r(S_t, A_t, S_{t+1})$. Similarly, we write $r(s, a) \triangleq \mathbb{E}_{S_{t+1} \sim p(\cdot|s,a)}[r(s, a, S_{t+1})]$ for brevity.

**Definition 2.** *(Single- and multi-step Bellman optimality operators). Given a model $m = (p, r)$, the single-step Bellman optimality operator, $\mathcal{T}^* : \mathcal{Q} \mapsto \mathcal{Q}$, is defined as:*

$$\mathcal{T}^* q(s, a) \triangleq \mathbb{E}_m \left[ r(s, a) + \gamma \max_{a' \in \mathcal{A}} q(S_{t+1}, a') | S_t = s, A_t = a \right]. \tag{6}$$

*The repeated application of $\mathcal{T}^*$ for $h$ steps gives rise to the $h$-step Bellman operator, denoted $(\mathcal{T}^*)^h q(s, a) \triangleq \underbrace{\mathcal{T}^* \cdots \mathcal{T}^*}_{h\text{-times}} q(s, a)$. Efroni et al. (2018) showed that $(\mathcal{T}^*)^h$ can be written as follows:*

$$(\mathcal{T}^*)^h q(s, a) \triangleq \mathbb{E}_m \left[ r(s, a) + \gamma \max_{a' \in \mathcal{A}} (\mathcal{T}^*)^{h-1} q(S_{t+1}, a') | S_t = s, A_t = a \right] \tag{7}$$

$$= \max_{\mu_1 \dots \mu_{h-1}} \mathbb{E}_m \left[ \sum_{k=0}^{h-1} \gamma^k r(S_{t+k}, \mu_k(S_{t+k})) + \gamma^h \max_{a' \in \mathcal{A}} q(S_{t+h}, a') | S_t = s, \mu_0(s) = a \right], \tag{8}$$

*where $\mu_i$ is any policy (in an arbitrary policy space) the agent could choose to deploy at time $i$.*

It is well known that $\mathcal{T}^*$ is a contraction mapping and its fixed-point (for any initial $q \in \mathcal{Q}$) is $\lim_{h \to \infty} (\mathcal{T}^*)^h q(s, a) = q^*(s, a)$ (Puterman, 2014).

We now introduce our first contribution: $h$-*step generalized policy improvement*, or $h$-GPI:

> **Definition 3.** *Let $\Pi = \{\pi_i\}_{i=1}^n$ be a set of policies and $m = (p, r)$ be a model. Then, given a horizon $h \geq 0$, the $h$-GPI policy on state $s$ is defined as*
>
> $$\pi^{h\text{-}GPI}(s) \in \arg\max_{a \in \mathcal{A}} (\mathcal{T}^*)^h \max_{\pi \in \Pi} q^\pi(s, a) \tag{9}$$
>
> $$= \arg\max_{a \in \mathcal{A}} \; \max_{\mu_1, \ldots, \mu_{h-1}} \mathbb{E}_m \left[ \underbrace{\sum_{k=0}^{h-1} \gamma^k r(S_{t+k}, \mu_k(S_{t+k}))}_{\textit{online planning}} + \gamma^h \underbrace{\max_{a' \in \mathcal{A}} \max_{\pi \in \Pi} q^\pi(S_{t+h}, a')}_{\textit{GPI}} \Big| \mu_0(S_t) = a \right] \tag{10}$$
>
> *where $\mu_i$ is any policy (in an arbitrary policy space) the agent could choose to deploy at time $i$.*

Intuitively, $h$-GPI identifies a policy that returns the best possible action, $a$, by first planning with model $m$ for $h$ steps and then using GPI to estimate the future returns achievable from all states reachable in $h$ steps. This is in contrast with standard GPI policies, which can only reason about the future returns achievable from the current state, $S_t$, if following a given policy in $\Pi$. $h$-GPI, by contrast, uses a model to reason over the decisions made at states within $h$ steps from $s$, as well as states, $S_{t+h}$, reachable after $h$ steps. This makes it possible for $h$-GPI to produce policies that exploit the model-free, zero-shot return estimates produced by GPI when evaluating states beyond a given horizon $h$; in particular, states that would otherwise not be considered by the standard GPI procedure when determining return-maximizing actions. Notice, furthermore, that $h$-GPI by construction interpolates between model-free GPI and fully model-based planning:

> $h$-GPI induces policies that interpolate between *(i)* standard model-free GPI (when $h = 0$) and *(ii)* fully model-based planning (when $h \to \infty$).

In practice, agents seldom have access to the true model, $m$, of an MDP, and instead rely on approximate models, $\hat{m}$. The application of the Bellman operator under an approximate model, $\hat{m}$, rather than the true MDP model, $m$, can be represented mathematically by replacing $m$ with $\hat{m}$ in Equation (6). This gives rise to a *model-dependent* Bellman optimality operator, which we call $\mathcal{T}^*_{\hat{m}}$. In what follows, we introduce Theorem 1, which extends the original GPI theorem (Barreto et al., 2017) and allows us to characterize the gains achievable by $h$-GPI as a function of approximation errors in both the agent's policy library ($\Pi$) and its (possibly learned) model $\hat{m}$. Notice that Theorem 1 generalizes the original GPI theorem, which is recovered when $h = 0$.

**Theorem 1.** *Let $\Pi = \{\pi_i\}_{i=1}^n$ be a set of policies, $\{\hat{q}^{\pi_i}\}_{i=1}^n$ be approximations of their respective action-value functions, and $\hat{m} = (\hat{p}, \hat{r})$ be an approximate model such that, for all $\pi_i \in \Pi$ and all $(s, a) \in \mathcal{S} \times \mathcal{A}$,*

$$|q^{\pi_i}(s, a) - \hat{q}^{\pi_i}(s, a)| \leq \epsilon, \quad \sum_{s'} |p(s'|s, a) - \hat{p}(s'|s, a)| \leq \epsilon_p, \text{ and } |r(s, a) - \hat{r}(s, a)| \leq \epsilon_r. \tag{11}$$

*Recall once again the definition of $h$-GPI (Definition 3):*

$$\pi^{h\text{-}GPI}(s) \in \arg\max_{a \in \mathcal{A}} (\mathcal{T}^*_{\hat{m}})^h \max_{\pi \in \Pi} \hat{q}^\pi(s, a). \tag{12}$$

*Then,*

$$q^{h\text{-}GPI}(s, a) \geq (\mathcal{T}^*)^h \max_{\pi \in \Pi} q^\pi(s, a) - \frac{2}{1 - \gamma} (\gamma^h \epsilon + c(\epsilon_r, \epsilon_p, h)) \tag{13}$$

$$\geq \max_{\pi \in \Pi} q^\pi(s, a) - \frac{2}{1 - \gamma} (\gamma^h \epsilon + c(\epsilon_r, \epsilon_p, h)), \tag{14}$$

*where $c(\epsilon_r, \epsilon_p, h) = \frac{1 - \gamma^h}{1 - \gamma}(\epsilon_r + \gamma \epsilon_p v^*_{max})$, and $v^*_{max} \triangleq |\max_{s,a} q^*(s, a)|$.*

> Theorem 1 characterizes the performance lower bound of $h$-GPI as a function of the number of planning steps, $h$, and the approximation errors in the agent's model and action-value functions (i.e., errors $\epsilon$, $\epsilon_p$, and $\epsilon_r$).

Note that since $(\mathcal{T}^*)^h \max_{\pi \in \Pi} q^\pi(s,a) \geq \max_{\pi \in \Pi} q^\pi(s,a)$, $h$-GPI's performance lower bound is strictly better than GPI's ($h = 0$) assuming no model approximation errors ($\epsilon_r = \epsilon_p = 0$). Furthermore, notice that as $h \to \infty$, $(\mathcal{T}^*)^h \max_{\pi \in \Pi} q^\pi(s,a)$ converges to $q^*$ and the approximation error term ($\gamma^h \epsilon$) in Equation (13) disappears. In other words, as $h$ increases, $h$-GPI's performance becomes arbitrarily less susceptible to sub-optimality in the agent's policy library. This implies that the planning horizon $h$ trades off between two conflicting objectives. On the one hand, increasing the planning horizon $h$ improves the performance of the $h$-GPI policy because the error term ($\epsilon$) associated with value function approximations errors becomes irrelevant. On the other hand, longer horizons worsen $h$-GPI's performance lower bound since they increase its dependency on errors arising from approximate models. Intuitively, for small values of $h$, the agent relies more heavily on the assumption that its estimates $\{\hat{q}^{\pi_i}\}_{i=1}^n$ are correct, given that these estimates are discounted by $\gamma^h$. For large values of $h$, by contrast, the approximate model error term, $c(\epsilon_r, \epsilon_p, h)$, increases, and the agent relies more heavily on the assumption that its (learned) model is approximately correct.

Next, we introduce a novel upper-bound on the discrepancy between the value, $q_{\mathbf{w}}^{h\text{-GPI}}$, of the $h$-GPI policy under a particular reward function, and the optimal action-value function, $q_{\mathbf{w}}^*$. Notice that, as in Theorem 1, we can recover the standard GPI bound when setting $h = 0$.[2]

**Theorem 2.** *Let $\Pi = \{\pi_i^*\}_{i=1}^n$ be a set of optimal policies with respect to reward weights $\{\mathbf{w}_i\}_{i=1}^n$ and $\mathbf{w}$ be arbitrary reward weights. Let $\hat{m} = (\hat{p}, \hat{r}_{\mathbf{w}})$ be an approximate model and $\{\hat{q}_{\mathbf{w}}^{\pi_i^*}\}_{i=1}^n$ be approximations to the action-value functions of policies in $\Pi$, under the reward function $r_{\mathbf{w}}$, such that for all $\pi_i \in \Pi$ and all $(s,a) \in \mathcal{S} \times \mathcal{A}$,*

$$|q_{\mathbf{w}}^{\pi_i^*}(s,a) - \hat{q}_{\mathbf{w}}^{\pi_i^*}(s,a)| \leq \epsilon, \quad \sum_{s'} |p(s'|s,a) - \hat{p}(s'|s,a)| \leq \epsilon_p, \text{ and } |r_{\mathbf{w}}(s,a) - \hat{r}_{\mathbf{w}}(s,a)| \leq \epsilon_r.$$

(15)

*We now extend the definition of $h$-GPI (Definition 3) to the case where this policy is defined under the assumption of an MPD with reward function $r_{\mathbf{w}}$:*

$$\pi^{h\text{-GPI}}(s; \mathbf{w}) \in \arg\max_{a \in \mathcal{A}} (\mathcal{T}_{\hat{m}}^*)^h \max_{\pi \in \Pi} \hat{q}_{\mathbf{w}}^\pi(s,a).$$

(16)

*Let $\phi_{max} \triangleq \max_{s,a} ||\phi(s,a)||$. Then, it follows that*

$$q_{\mathbf{w}}^*(s,a) - q_{\mathbf{w}}^{h\text{-GPI}}(s,a) \leq \frac{2}{1-\gamma}(\phi_{max} \min_i ||\mathbf{w} - \mathbf{w}_i|| + \gamma^h \epsilon + c(\epsilon_r, \epsilon_p, h)),$$

(17)

*where $c(\epsilon_r, \epsilon_p, h) = \frac{1-\gamma^h}{1-\gamma}(\epsilon_r + \gamma \epsilon_p v_{max}^*)$.*

> Theorem 2 precisely characterizes the optimality gap (i.e., the maximum difference) between the action-value function induced by $h$-GPI ($q_{\mathbf{w}}^{h\text{-GPI}}(s,a)$) and the optimal action-value function ($q_{\mathbf{w}}^*(s,a)$), as a function of (i) the reward weights $\{\mathbf{w}_i\}_{i=1}^n$ for which the policies in the agent's library, $\Pi$, are optimal; (ii) approximation errors in action-value functions $\{\hat{q}_{\mathbf{w}}^{\pi_i^*}\}_{i=1}^n$; and (iii) approximation errors in the model $\hat{m} = (\hat{p}, \hat{r}_{\mathbf{w}})$.

### 3.1 $h$-GPI under (learned) approximate models for zero-shot transfer

Recall that our goal is to perform zero-shot policy transfer to solve any tasks $M \in \mathcal{M}^\phi$ (Equation (1)). In order to employ $h$-GPI on *any* given reward function of the form $r_{\mathbf{w}}(S_t, A_t, S_{t+1}) = \phi_t \cdot \mathbf{w}$, we propose learning a model, $\hat{m} = (\hat{p}, \hat{\phi})$, that approximates both the state transition function, $p$, and *reward features*, $\phi$. The key insight is that this model implicitly induces the space of all MDP models $m = (p, r_{\mathbf{w}})$, based on (arbitrary) reward weights $\mathbf{w} \in \mathbb{R}^d$. This allows employing RL model-based methods to multi-task settings without the need to re-train the model when solving new tasks defined by different reward functions. In Algorithm 1, we present a high-level description of how the $h$-GPI policy $\pi^{h\text{-GPI}}(s, \mathbf{w})$ can be computed given a model $\hat{m} = (\hat{p}, \hat{\phi})$, SFs $\{\hat{\psi}^{\pi_i}\}_{i=1}^n$, and horizon $h \geq 0$.

**Tabular setting.** In the tabular case, models can be learned via maximum likelihood estimates: $\hat{m}(s', \phi | s, a) \approx \frac{N(s, a, \phi, s')}{N(s,a)}$, where $N(s, a, \phi, s')$ is the number of times the agent experienced the

---

[2]The proofs of Theorem 1 and Theorem 2 can be found in Appendix A.

---

**Algorithm 1:** $h$-GPI with Successor Features

---

**Input :** Model $\hat{m} = (\hat{p}, \hat{\phi})$, SFs $\{\hat{\psi}^{\pi_i}\}_{i=1}^n$, planning horizon $h \geq 0$, state $s$, reward weights $\mathbf{w}$.

1 **for** *action* $a \in \mathcal{A}$ **do**

2     Let $S_t = s$, $\mu_0(s) = a$

3     Compute $(\mathcal{T}_{\hat{m}}^*)^h \max_{\pi \in \Pi} \hat{q}_{\mathbf{w}}^\pi(s, a) \leftarrow$

$$\max_{\mu_1 \dots \mu_{h-1}} \mathbb{E}_{\hat{m}} \left[ \sum_{k=0}^{h-1} \gamma^k \hat{\phi}_{t+k}(\hat{S}_{t+k}, \mu_k(\hat{S}_{t+k})) \cdot \mathbf{w} + \gamma^h \max_{a' \in \mathcal{A}} \max_{\pi \in \Pi} \hat{\psi}^\pi(\hat{S}_{t+h}, a') \cdot \mathbf{w} \right]$$

4 **Return:** $\pi^{h\text{-GPI}}(s; \mathbf{w}) \in \arg\max_{a \in \mathcal{A}} (\mathcal{T}_{\hat{m}}^*)^h \max_{\pi_i \in \Pi} \hat{q}_{\mathbf{w}}^{\pi_i}(s, a)$

---

transition $(s, a, \phi, s')$ and $N(s, a)$ is the number of times the agent selected action $a$ in state $s$. To efficiently compute the action given by the policy $\pi^{h\text{-GPI}}(s, \mathbf{w})$ (line 3 of Algorithm 1), we extend the Forward-Backward Dynamic Programming (FB-DP) algorithm (Efroni et al., 2020) to the discounted-SFs setting, in order to compute $h$-lookahead policies in real-time. The corresponding pseudocode can be found in Appendix B. Notably, given a state $s$, FB-DP time complexity is $O(N|\mathcal{A}||\mathcal{S}_h^{\text{tot}}|)$, where $N$ is the maximal number of accessible states in one step (maximal "nearest neighbors" from any state), and $\mathcal{S}_h^{\text{tot}}$ is total reachable states in $h$ time steps from state $s$. Note that while we chose to extend FB-DP due to its efficiency and closed-form formulation, other planning techniques could have been used (e.g., Monte Carlo tree search (Tesauro and Galperin, 1996; Silver et al., 2017)).

**Continuous-states setting.** In the continuous-state setting, we extend the class of models composed of ensembles of probabilistic neural networks (Chua et al., 2018) to the SFs setting. These models are used in state-of-the-art single-task model-based RL algorithms (Janner et al., 2019; Yu et al., 2021). The learned model $\hat{m}_\varphi(s', \phi|s, a)$, parameterized by $\varphi$, is composed of an ensemble of $n$ neural networks, $\{\hat{m}_{\varphi_i}\}_{i=1}^n$, each of which outputs the mean and diagonal covariance matrix of a multivariate Gaussian distribution: $m_{\varphi_i}(S_{t+1}, \phi_t \mid S_t, A_t) = \mathcal{N}(\mu_{\varphi_i}(S_t, A_t), \Sigma_{\varphi_i}(S_t, A_t))$. Each model in the ensemble is trained in parallel to minimize the following negative log-likelihood loss function, using different bootstraps of experiences in a buffer $\mathcal{B} = \{(S_t, A_t, \phi_t, S_{t+1})|t \geq 0\}$: $\mathcal{L}(\varphi) = \mathbb{E}_{(S_t, A_t, \phi_t, S_{t+1}) \sim \mathcal{B}}[-\log m_\varphi(S_{t+1}, \phi_t|S_t, A_t)]$. In practice, we use as $\mathcal{B}$ the buffer with experiences the agent collected while training the SFs for the training tasks. In order to compute the $h$-GPI policy in this setting, we approximate the expectation over next states by averaging over the predictions of the components of the ensemble (Chua et al., 2018).

We employ *universal successor features approximators* (USFAs) (Borsa et al., 2019) to learn SFs in the function approximation setting. Given sufficiently expressive USFAs, we can evaluate the value function of *any* policy $\pi_{\mathbf{z}}$ (optimal for task $\mathbf{z}$) in *any* task $\mathbf{w} \in \mathbb{R}^d$ by simply computing $\hat{q}_{\mathbf{w}}^{\pi_{\mathbf{z}}}(s, a) \approx \hat{\psi}(s, a, \mathbf{z}) \cdot \mathbf{w}$. Furthermore, when using a USFA to generalize to a new task $\mathbf{w}'$, GPI (Equation 5) becomes: $\pi^{\text{GPI}}(s; \mathbf{w}) \in \arg\max_{a \in \mathcal{A}} \max_{\mathbf{w} \in \mathcal{M}} \hat{\psi}(s, a, \mathbf{w}) \cdot \mathbf{w}'$, where $\mathcal{M}$ is typically a set of weight vectors used when training the USFA. Notably, we only require a single USFA $\hat{\psi}$ and a single model $\hat{m}$ to perform $h$-GPI given any $M \in \mathcal{M}_\phi$, in contrast to other approaches that require learning a complex model for each policy in the library (Thakoor et al., 2022).

Notice that although we assume that during training the agent can observe and add to its buffer $\mathcal{B}$ the features $\phi_t$ at each time step $t$, our method could also be applied to the setting where the features are learned beforehand in a pre-training phase (Hansen et al., 2020; Carvalho et al., 2023). Learning appropriate reward features $\phi$ is an important but orthogonal problem to the one tackled in this paper.

## 4 Experiments

We conduct tabular and deep RL experiments in three different domains to evaluate the effectiveness of $h$-GPI as a method for zero-shot policy transfer.

**Environments.** First, we consider the tabular FourRoom domain (Barreto et al., 2017). To make this domain more challenging, we made its transition function stochastic by including a 10% chance of the agent moving to a random direction after taking any action. In FourRoom, the reward features $\phi_t \in \mathbb{R}^3$ correspond to binary vectors indicating whether the agent collected each of the three types of objects in the map. The second domain is Reacher (Alegre et al., 2022a), which consists of a robotic arm that can apply torque to each of its two joints. The features $\phi_t \in \mathbb{R}^4$ are proportional

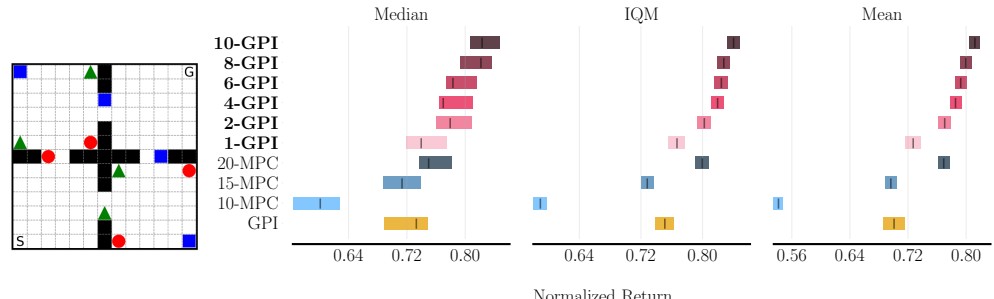

Figure 2: Median, IQM, and mean normalized returns over a set of test tasks in the stochastic FourRoom domain. $h$-GPI performance significantly improves as $h$ increases.

to the distance of the tip of the arm to four different targets. Finally, we extend the FetchPush domain (Plappert et al., 2018), which consists of a fetch robotic arm that must move a block to a target position on top of a table by pushing the block with its gripper. The reward features $\phi_t \in \mathbb{R}^4$ are the negative distances of the block to each of the four target locations. The action space was discretized in the same way as usually done in Reacher. Importantly, the state space of this domain is high-dimensional ($\mathcal{S} \subset \mathbb{R}^{19}$) and its dynamics are significantly more complex than Reacher's. A more detailed description of the domains can be found in Appendix B.

**Baselines.** We compare $h$-GPI with standard GPI (Barreto et al., 2017), which is equivalent to $h$-GPI when $h = 0$. To showcase the importance of combining model-based planning with model-free action-value estimates, we include an SF-based MPC approach, which is equivalent to $h$-GPI but without GPI bootstrapping at time step $h$. That is, given a task $\mathbf{w}$, it selects the first action from a sequence of actions that maximizes $\mathbb{E}_{\hat{m}}[\sum_{k=0}^{h-1} \gamma^k \hat{\phi}_{t+k} \cdot \mathbf{w}]$, and re-plans at every time step. In the function approximation case with USFAs, we also compare with Constrained GPI (CGPI) (Kim et al., 2022). CGPI is a state-of-the-art method that computes lower and upper bounds for the optimal value on new tasks and uses them to constrain value approximation errors when following the GPI policy.

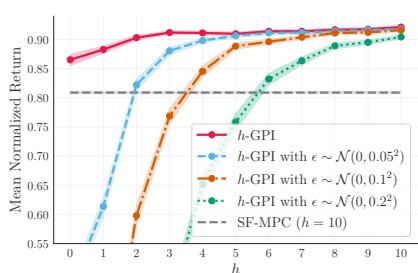

Figure 3: $h$-GPI, under various levels of errors in the SFs, as a function of $h$.

Based on the theoretical properties introduced in Section 3, we expect the following hypotheses to hold. **H1:** $h$-GPI performance better generalizes to unseen test tasks than standard GPI and, in general, improves as we increase $h$. **H2:** $h$-GPI is more robust to approximation errors in the learned SFs than GPI, and such errors have a decreasing impact as $h$ increases. Additionally, we investigate whether **H3:** $h$-GPI can be scaled up to solve problems with high-dimensional state spaces. In each experiment, we report the median, interquartile mean (IQM), and mean normalized returns with their 95% stratified bootstrap confidence intervals (Agarwal et al., 2021), which are obtained by evaluating agents trained with 20 different random seeds on a set of test tasks. For fairness of comparison, we used the same agents for evaluating each method. We follow previous works (Borsa et al., 2019; Kim et al., 2022) and use as training tasks the weight vectors that form the standard basis of $\mathbb{R}^d$ in all three domains. In FourRoom, we use 32 weights vectors equally spaced from the weight simplex $\{\mathbf{w} \mid \sum_{i=1}^d w_i = 1, w_i \geq 0\}$ as test tasks. For Reacher and FetchPush, we follow Kim et al. (2022) and use weight vectors defined by $\{-1, 1\}^d$ as test tasks. Notice that this results in many tasks with negative weights, which are significantly different than the training tasks.

In Figure 2, we can observe that $h$-GPI is able to outperform GPI even with small planning horizons (see, e.g., 1-GPI). As $h$ increases, the performance on the test tasks consistently increases. This is in accordance with Theorem 1 and **H1**. Notice that even with a small planning horizon of $h = 2$, $h$-GPI is capable of matching the performance of SF-MPC even when SF-MPC is allowed to plan for *ten times* longer (i.e., $h = 20$).

Next, to investigate **H2**, we evaluate $h$-GPI after artificially adding Gaussian noise with different standard deviations to the learned SFs of each of the agent's policies for every state and action. We can notice (in Figure 3) that standard GPI ($h = 0$) catastrophically fails under all levels of

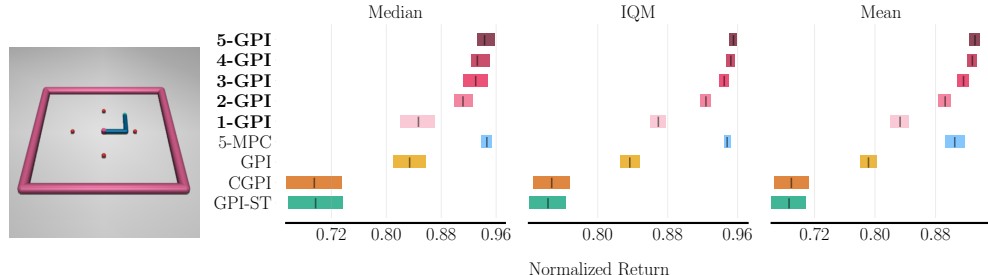

Figure 4: Median, IQM, and mean normalized returns over test tasks in the Reacher domain.

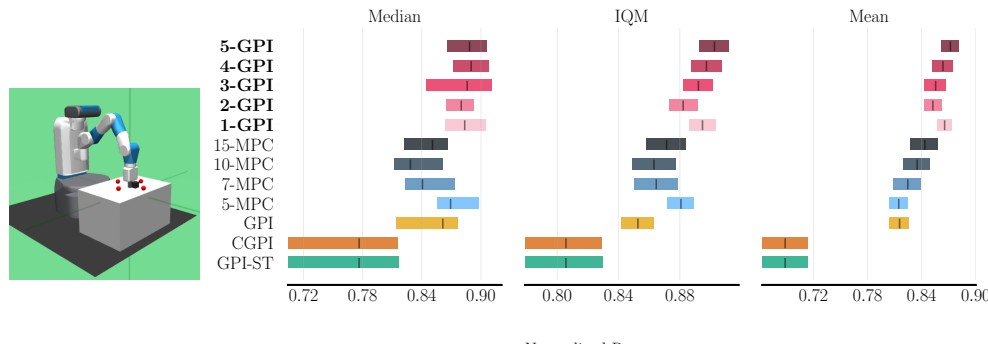

Figure 5: Median, IQM, and mean normalized returns over test tasks in the FetchPush domain.

approximation errors. In all cases, $h$-GPI can closely approximate the same performance levels as those resulting from settings with *no* added noise. To do so, it suffices to use a sufficiently large $h$. In other words, $h$-GPI successfully and effectively identifies policies that are robust to noise due to its ability to combine model-free GPI generalization and planning under approximate/imperfect models for $h$ steps. This is consistent with our theoretical results that show that value function approximation errors impact the performance's lower bound by a factor of $\gamma^h$.

In Figures 4 and 5, we show the results in the Reacher and FetchPush domains, respectively. Given a test task $\mathbf{w}'$, let GPI-ST be the policy that uses both the source/training (S) and target (T) tasks, $\mathcal{M} \cup \{\mathbf{w}'\}$, as input to the USFA when performing GPI. Similarly, we refer to GPI as the policy that performs GPI only over the training tasks $\mathcal{M}$. The CGPI policy was unable to significantly improve performance over GPI-ST, which we hypothesize may be due to USFA errors in the training tasks, or due to overly small minimum rewards used in its upper-bound. Both are known limitations of CGPI (Kim et al., 2022). In both domains with function approximation using USFAs, $h$-GPI outperforms the competing methods, which supports **H1** and **H3**. In the FetchPush domain, in particular, SF-MPC is allowed to plan for up to $h=15$ steps; that is, *three times* longer than the maximum horizon we allow $h$-GPI to plan. Again, $h$-GPI consistently outperforms it (and other baselines) even when allowed to plan for significantly fewer steps. Interestingly, in the FetchPush domain, $h$-GPI's performance slightly decreases for intermediate values of $h$, which suggests that its policy is being affected by model approximation errors. Even so, the mean returns achieved, for all values of $h$ are higher than the ones achieved by GPI. This demonstrates the robustness of $h$-GPI to approximate models of the environment and function approximation errors.

## 5 Discussion and related work

We now discuss the works most closely related to $h$-GPI. Further discussions can be found in Appendix C.

**GPI and SFs.** Previous works have extended GPI to safe RL (Gimelfarb et al., 2021; Feng et al., 2023), maximum-entropy RL (Hunt et al., 2019), unsupervised RL (Hansen et al., 2020), and hierarchical RL (e.g., via the options framework) (Barreto et al., 2019; Machado et al., 2023). Recently, Thakoor et al. (2022) introduced Geometric GPI (GGPI). GGPI uses geometric horizon models (GHMs) to learn the discounted future state-visitation distribution induced by particular policies to rapidly

evaluate policies under a given, known reward function. The authors show that performing GPI over a particular type of non-stationary policy produces behaviors that outperform those in the agent's library policy. $h$-GPI, by contrast, learns a different type of model: an environment model, which is used to perform planning—i.e., action selection—rather than policy evaluation. Additionally, GGPI requires learning separate GHMs for each policy in the library, whereas $h$-GPI can operate with a single model that predicts the next reward features, alongside a single USFA. Secondly, GGPI assumes that the reward function is known *a priori*, while we exploit SFs to generalize over all linear rewards given reward features. Bagot et al. (2023) introduced GPI-Tree Search (GPI-TS), which is closely related to $h$-GPI. GPI-TS uses GPI bootstrapping as backup value estimates at the leaf nodes of Monte Carlo tree search. However, GPI-TS does not employ SFs and was only designed to tackle single-task settings. Moreover, it assumes an oracle model of the environment. We, by contrast, exploit learned models to perform zero-shot transfer over multiple tasks and show how approximation errors in the model affect the performance of the $h$-GPI policy (see Theorems 1 and 2). Kim et al. (2022) introduced Constrained GPI (CGPI), which uses lower and upper bounds of the value of a new task to constrain the action-value estimates used when selecting the GPI policy's action. Although CGPI is able to deal with generalization errors in the values given by USFAs for target tasks, it is sensitive to errors in the value estimates for the training tasks. $h$-GPI, by contrast, can deal with approximation errors in the value functions of the source/base policies. Other works have studied methods for constructing a policy library for use with GPI (Zahavy et al., 2021; Nemecek and Parr, 2021; Alver and Precup, 2022; Alegre et al., 2022a) and for learning different SF-based representations (Lehnert and Littman, 2020; Touati and Ollivier, 2021). These methods solve important orthogonal problems and could potentially be combined with $h$-GPI.

**Multi-step RL algorithms.** Multi-step RL methods were extensively studied for policy evaluation, both in the model-free (Hessel et al., 2018; van Hasselt et al., 2018) and the model-based settings (Yao et al., 2009; Janner et al., 2019). Model value expansion algorithms (Feinberg et al., 2018; Buckman et al., 2018; Abbas et al., 2020) are a significant example of the latter. In this work, by contrast, we introduce a multi-step method for policy improvement in transfer learning settings. GX-Chen et al. (2022) introduced the $\eta$-return mixture, a new backup target for better credit assignment, which combines bootstrapping with standard value estimates and SFs, as a function of a parameter $\eta$. Efroni et al. (2018) studied multi-step greedy versions of the well-known dynamic programming (DP) policy iteration and value iteration algorithms (Bertsekas and Tsitsiklis, 1996), and Efroni et al. (2020) proposed a multi-step greedy real-time DP algorithm that replaces 1-step greedy policies with a $h$-step lookahead policy. Our work is also related to the techniques introduced by Sikchi et al. (2022) and Hansen et al. (2022), which combine planning and bootstrapping with learned value estimates. However, unlike $h$-GPI, these methods do not address the multi-policy and zero-shot transfer settings. We also note that there exists neuroscientific evidence that multi-step planning occurs in the brains of humans and other animals (Miller and Venditto, 2021). Finally, we believe that our work also shares similarities with the investigation performed by Tomov et al. (2021) on how model-free and model-based mechanisms are combined in human behavior.

# 6 Conclusions

We introduced $h$-GPI, a multi-step extension of GPI that interpolates between standard model-free GPI and fully model-based planning. Through novel theoretical and empirical results, we showed that $h$-GPI effectively exploits approximate models to solve novel tasks in a zero-shot manner. Notably, in our experiments, and consistent with the introduced theorems, we showed that $h$-GPI is less susceptible to value approximation errors and that it outperforms standard GPI and state-of-the-art competing baselines. Our findings hold even in high-dimensional problems where imperfect learned models are used. These results, combined with our method's strong formal guarantees, indicate that $h$-GPI is an important first step towards bridging the gap between model-free GPI-based methods and model-based planning algorithms—while also being robust to approximation errors.

**Limitations and future work.** The main limitation of $h$-GPI is that it can introduce computational overhead for large values of $h$. In future work, this can be tackled by designing heuristics for use when performing online planning, or via principled methods to dynamically select the best value of $h$. Another interesting direction is to integrate uncertainty estimation techniques (e.g., *implicit value ensemble* (Filos et al., 2022)) into the $h$-GPI policy to further reduce the impact of high model or value function approximation errors. Finally, regarding potential direct negative societal impacts of this work, we do not anticipate any.

## Acknowledgments and disclosure of funding

This study was financed in part by the Coordenação de Aperfeiçoamento de Pessoal de Nível Superior - Brasil (CAPES) - Finance Code 001; CNPq (Grants 140500/2021-9, 304932/2021-3); FAPESP/MCTI/CGI (Grant 2020/05165-1); the Research Foundation Flanders (FWO) [G062819N]; the AI Research Program from the Flemish Government (Belgium); and the Francqui Foundation.

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

# Appendix

## A  Proofs of theoretical results

We start by defining the Bellman optimality operator, $\mathcal{T}^*$, and its multi-step variant, when applied to action-value functions $q \in \mathcal{Q}$, where $\mathcal{Q}$ is the space of action-value functions. To simplify notation, given the (true) model, $m \triangleq (p, r)$, of the underlying MDP, we denote $\mathbb{E}_m[\cdot]$ as the expectation operator with respect to $S_{t+1} \sim p(.|S_t, A_t)$ and $R_t \triangleq r(S_t, A_t, S_{t+1})$. Similarly, we write $r(s, a) \triangleq \mathbb{E}_{S_{t+1} \sim p(\cdot|s,a)}[r(s, a, S_{t+1})]$ for brevity.

**Definition** (2). *(Single- and multi-step Bellman optimality operators). Given an MDP's model $m = (p, r)$, the single-step Bellman optimality operator, $\mathcal{T}^* : \mathcal{Q} \mapsto \mathcal{Q}$, is defined as:*

$$\mathcal{T}^* q(s, a) \triangleq \mathbb{E}_m[r(s, a) + \gamma \max_{a' \in \mathcal{A}} q(S_{t+1}, a')|S_t = s, A_t = a]. \tag{18}$$

*The repeated application of $\mathcal{T}^*$ for $h$ steps gives rise to the $h$-step Bellman operator, denoted as $(\mathcal{T}^*)^h q(s, a) \triangleq \underbrace{\mathcal{T}^* \cdots \mathcal{T}^*}_{h\text{-times}} q(s, a)$. Efroni et al. (2018) showed that $(\mathcal{T}^*)^h$ can be written as follows:*

$$(\mathcal{T}^*)^h q(s, a) \triangleq \mathbb{E}_m \left[ r(s, a) + \gamma \max_{a' \in \mathcal{A}} (\mathcal{T}^*)^{h-1} q(S_{t+1}, a')|S_t = s, A_t = a \right] \tag{19}$$

$$= \max_{\mu_1 \dots \mu_{h-1}} \mathbb{E}_m \left[ \sum_{k=0}^{h-1} \gamma^k r(S_{t+k}, \mu_k(S_{t+k})) + \gamma^h \max_{a' \in \mathcal{A}} q(S_{t+h}, a')|S_t = s, \mu_0(s) = a \right], \tag{20}$$

*where $\mu_i$ is any policy (in an arbitrary policy space) the agent could choose to deploy at time $i$.*

**Definition 4.** *(Bellman evaluation operator). Given an MDP's model $m = (p, r)$ and a policy $\pi$, the single-step Bellman evaluation operator, $\mathcal{T}^\pi : \mathcal{Q} \mapsto \mathcal{Q}$, is defined as:*

$$\mathcal{T}^\pi q(s, a) \triangleq \mathbb{E}_m \left[ r(s, a) + \gamma q(S_{t+1}, \pi(S_{t+1}))|S_t = s, A_t = a \right]. \tag{21}$$

It is well known that $\mathcal{T}^*$ and $\mathcal{T}^\pi$ are contraction mappings and their fixed-point (for any initial $q \in \mathcal{Q}$) are $\lim_{h \to \infty} (\mathcal{T}^*)^h q(s, a) = q^*(s, a)$ and $\lim_{h \to \infty} (\mathcal{T}^\pi)^h q(s, a) = q^\pi(s, a)$, respectively (Puterman, 2014).

Notice that in the definitions above, since $m$ is the (true) model of a given MDP, we chose to omit $m$ when writing the corresponding Bellman operators to make the notation more succinct. Formally, though, one could alternatively denote these operators as $\mathcal{T}^*_m$ and $\mathcal{T}^\pi_m$, respectively. In what follows, we always omit $m$ whenever the corresponding operators are defined with respect to the true model of an MDP.

The Bellman optimality operator induced by *an approximate model*, $\hat{m} = (\hat{p}, \hat{r})$, of the true underlying transition and reward functions, $p$ and $r$, of a given MDP, is defined as:

$$\mathcal{T}^*_{\hat{m}} q(s, a) \triangleq \mathbb{E}_{\hat{m}}[\hat{r}(s, a) + \gamma \max_{a' \in \mathcal{A}} q(S_{t+1}, a')|S_t = s, A_t = a] \tag{22}$$

$$= \hat{r}(s, a) + \gamma \sum_{s'} \hat{p}(s'|s, a) \max_{a' \in \mathcal{A}} q(s', a'). \tag{23}$$

Now, recall our definition of $h$-*step generalized policy improvement*, or $h$-GPI:

> **Definition** (3). *Let $\Pi = \{\pi_i\}_{i=1}^n$ be a set of policies and $m = (p, r)$ be a model. Then, given a horizon $h \geq 0$, the $h$-GPI policy on state $s$ is defined as*
>
> $$\pi^{h\text{-}GPI}(s) \in \arg\max_{a \in \mathcal{A}} (\mathcal{T}_m^*)^h \max_{\pi \in \Pi} q^\pi(s, a) \tag{24}$$
>
> $$= \arg\max_{a \in \mathcal{A}} \max_{\mu_1, \ldots, \mu_{h-1}} \mathbb{E}_m \left[ \underbrace{\sum_{k=0}^{h-1} \gamma^k r(S_{t+k}, \mu_k(S_{t+k}))}_{\textit{online planning}} + \gamma^h \underbrace{\max_{a' \in \mathcal{A}} \max_{\pi \in \Pi} q^\pi(S_{t+h}, a')}_{\textit{GPI}} \, | \mu_0(S_t) = a \right]$$
>
> $$\tag{25}$$
>
> *where $\mu_i$ is any policy (in an arbitrary policy space) the agent could choose to deploy at time $i$.*

Intuitively, an $h$-GPI policy identifies the best possible action, $A_t$, when in state $S_t$, by first planning with model $m$ for $h$ steps and then using GPI to estimate the future returns achievable from all states reachable in $h$ steps. This is in contrast with standard GPI policies, which can only reason about the future returns achievable from the current state, $S_t$, if following a given policy in $\Pi$. $h$-GPI, by contrast, uses a model to reason over the decisions made at states within $h$ steps from $S_t$, as well as states $S_{t+h}$ reachable after $h$ steps. This makes it possible for $h$-GPI to produce policies that exploit the model-free, zero-shot return estimates produced by GPI when evaluating states beyond a given horizon $h$; in particular, states that would otherwise not be considered by the standard GPI procedure when determining return-maximizing actions. Notice, furthermore, that $h$-GPI—by construction—interpolates between model-free GPI and fully model-based planning:

> $h$-GPI induces policies that interpolate between *(i)* standard model-free GPI (when $h = 0$) and *(ii)* fully model-based planning (when $h \to \infty$).

Next, we introduce a few lemmas that will subsequently allow us to *(i)* precisely characterize the performance lower bound of an $h$-GPI policy; *(ii)* show that it is strictly better than GPI's performance lower bound; *(iii)* prove that as $h$ increases, $h$-GPI's performance becomes arbitrarily less susceptible to sub-optimality in the agent's policy library.

The following lemma characterizes the loss introduced by applying the multi-step Bellman optimality operator, $(\mathcal{T}^*)^h$, to an approximate action-value function, $\hat{q}^\pi(s, a)$.

**Lemma 1.** *Let $\Pi = \{\pi_i\}_{i=1}^n$ be a set of policies, and $\{\hat{q}^{\pi_i}\}_{i=1}^n$ be approximations to their respective action-value functions such that, for all $\pi_i \in \Pi$ and for all $(s, a) \in \mathcal{S} \times \mathcal{A}$,*

$$|q^{\pi_i}(s, a) - \hat{q}^{\pi_i}(s, a)| \leq \epsilon. \tag{26}$$

*Then,*

$$(\mathcal{T}^*)^h \max_{\pi \in \Pi} \hat{q}^\pi(s, a) \geq (\mathcal{T}^*)^h \max_{\pi \in \Pi} q^\pi(s, a) - \gamma^h \epsilon. \tag{27}$$

*Proof.*

$$(\mathcal{T}^*)^h \max_{\pi \in \Pi} \hat{q}^\pi(s, a) = \max_{\mu_1 \ldots \mu_{h-1}} \mathbb{E} \left[ \sum_{k=0}^{h-1} \gamma^k r(S_{t+k}, \mu_k(S_{t+k})) + \gamma^h \max_{a' \in \mathcal{A}} \max_{\pi \in \Pi} \hat{q}^\pi(S_{t+h}, a') | S_t = s, \mu_0(s) = a \right] \tag{28}$$

$$\geq \max_{\mu_1 \ldots \mu_{h-1}} \mathbb{E} \left[ \sum_{k=0}^{h-1} \gamma^k r(S_{t+k}, \mu_k(S_{t+k})) + \gamma^h \max_{a' \in \mathcal{A}} \max_{\pi \in \Pi} (q^\pi(S_{t+h}, a') - \epsilon) | S_t = s, \mu_0(s) = a \right] \tag{29}$$

$$= \max_{\mu_1 \ldots \mu_{h-1}} \mathbb{E} \left[ \sum_{k=0}^{h-1} \gamma^k r(S_{t+k}, \mu_k(S_{t+k})) + \gamma^h \max_{a' \in \mathcal{A}} \max_{\pi \in \Pi} q^\pi(S_{t+h}, a') | S_t = s, \mu_0(s) = a \right] - \gamma^h \epsilon \tag{30}$$

$$= (\mathcal{T}^*)^h \max_{\pi \in \Pi} q^\pi(s, a) - \gamma^h \epsilon. \tag{31}$$

Above, (28) and (31) are due to the definition of $(\mathcal{T}^*)^h$ introduced in Equation (20). $\qquad\square$

The next lemma characterizes the loss incurred by applying the Bellman optimality operator induced by an approximate model $\hat{m}$, $(\mathcal{T}_{\hat{m}}^*)^h$, to an action-value function, $q^\pi$, of a policy $\pi$.

**Lemma 2.** *Let $\hat{m} = (\hat{p}, \hat{r})$ be an approximate model such that, for all $(s,a) \in \mathcal{S} \times \mathcal{A}$,*

$$\sum_{s'} |p(s'|s,a) - \hat{p}(s'|s,a)| \leq \epsilon_p \quad and \quad |r(s,a) - \hat{r}(s,a)| \leq \epsilon_r. \tag{32}$$

*Let $v_{max}^* \triangleq |\max_{s,a} q^*(s,a)|$. Then, for any policy $\pi$ and $h \geq 0$,*

$$|(\mathcal{T}^*)^h q^\pi(s,a) - (\mathcal{T}_{\hat{m}}^*)^h q^\pi(s,a)| \leq c(\epsilon_r, \epsilon_p, h), \tag{33}$$

*where*

$$c(\epsilon_r, \epsilon_p, h) = \frac{1 - \gamma^h}{1 - \gamma}(\epsilon_r + \gamma \epsilon_p v_{max}^*). \tag{34}$$

*Proof.*

$$\left| (\mathcal{T}^*)^h q^\pi(s,a) - (\mathcal{T}_{\hat{m}}^*)^h q^\pi(s,a) \right| =$$

$$\left| r(s,a) + \gamma \sum_{s'} p(s'|s,a) \max_{a' \in \mathcal{A}} (\mathcal{T}^*)^{h-1} q^\pi(s',a') - \hat{r}(s,a) - \gamma \sum_{s'} \hat{p}(s'|s,a) \max_{a' \in \mathcal{A}} (\mathcal{T}_{\hat{m}}^*)^{h-1} q^\pi(s',a') \right| \tag{35}$$

$$\leq \epsilon_r + \gamma \left| \sum_{s'} p(s'|s,a) \max_{a' \in \mathcal{A}} (\mathcal{T}^*)^{h-1} q^\pi(s',a') - \sum_{s'} \hat{p}(s'|s,a) \max_{a' \in \mathcal{A}} (\mathcal{T}_{\hat{m}}^*)^{h-1} q^\pi(s',a') \right| \tag{36}$$

$$= \epsilon_r + \gamma \left| \sum_{s'} p(s'|s,a) \max_{a' \in \mathcal{A}} (\mathcal{T}^*)^{h-1} q^\pi(s',a') + \sum_{s'} \hat{p}(s'|s,a) \max_{a' \in \mathcal{A}} (\mathcal{T}^*)^{h-1} q^\pi(s',a') \right.$$

$$\left. - \sum_{s'} \hat{p}(s'|s,a) \max_{a' \in \mathcal{A}} (\mathcal{T}^*)^{h-1} q^\pi(s',a') - \sum_{s'} \hat{p}(s'|s,a) \max_{a' \in \mathcal{A}} (\mathcal{T}_{\hat{m}}^*)^{h-1} q^\pi(s',a') \right| \tag{37}$$

$$= \epsilon_r + \gamma \left| \sum_{s'} \hat{p}(s'|s,a) \left( \max_{a' \in \mathcal{A}} (\mathcal{T}^*)^{h-1} q^\pi(s',a') - \max_{a' \in \mathcal{A}} (\mathcal{T}_{\hat{m}}^*)^{h-1} q^\pi(s',a') \right) \right.$$

$$\left. + \sum_{s'} (p(s'|s,a) - \hat{p}(s'|s,a)) \max_{a' \in \mathcal{A}} (\mathcal{T}^*)^{h-1} q^\pi(s',a') \right| \tag{38}$$

$$\leq \epsilon_r + \gamma \max_{s',a'} \left| (\mathcal{T}^*)^{h-1} q^\pi(s',a') - (\mathcal{T}_{\hat{m}}^*)^{h-1} q^\pi(s',a') \right|$$

$$+ \gamma \sum_{s'} \left| p(s'|s,a) - \hat{p}(s'|s,a) \right| \left| \max_{a' \in \mathcal{A}} (\mathcal{T}^*)^{h-1} q^\pi(s',a') \right| \tag{39}$$

$$\leq \epsilon_r + \gamma \max_{s',a'} \left| (\mathcal{T}^*)^{h-1} q^\pi(s',a') - (\mathcal{T}_{\hat{m}}^*)^{h-1} q^\pi(s',a') \right| + \gamma \epsilon_p v_{max}^*. \tag{40}$$

In the proof above, all terms shown in blue refer to approximate quantities—such as an approximate model of the environment or an approximate transition function. In the proof above, (35) follows from the recursive definition of $(\mathcal{T}^*)^h$ showed in Equation (19). In (37), we add and subtract $\sum_{s'} \hat{p}(s'|s,a) \max_{a' \in \mathcal{A}} (\mathcal{T}^*)^{h-1} q^\pi(s',a')$. (39) is obtained due to the property that $\sum_{s'} \hat{p}(s'|s,a) f(s') \leq \max_{s'} f(s')$, for any function $f$. Finally, (40) is due to $(\mathcal{T}^*)^{h-1} q^\pi(s',a') \leq v_{max}^*$.

We now show how to rewrite the inequality (40) recursively. Let $\Delta_h(s,a) \triangleq |(\mathcal{T}^*)^h q^\pi(s,a) - (\mathcal{T}_{\hat{m}}^*)^h q^\pi(s,a)|$, for all $(s,a) \in \mathcal{S} \times \mathcal{A}$. Then, by replacing the definition of $\Delta_h$ in (40), we obtain:

$$\Delta_h(s,a) \leq \epsilon_r + \gamma \max_{s',a'} \Delta_{h-1}(s',a') + \gamma \epsilon_p v_{max}^*. \tag{41}$$

Because inequality (41) holds for any $(s,a)$, it also holds for the maximizer. Thus, we can omit $(s,a)$, and define $\Delta_h \triangleq \max_{s,a} \Delta_h(s,a)$. The inequality (41) then becomes:

$$\Delta_h \leq \epsilon_r + \gamma \Delta_{h-1} + \gamma \epsilon_p v_{max}^*. \tag{42}$$

Now we use the fact that, by definition, $\Delta_0(s, a) = |q^\pi(s, a) - q^\pi(s, a)| = 0$, to recursively expand inequality (42):

$$\Delta_h \leq \sum_{k=0}^{h-1} \gamma^k (\epsilon_r + \gamma \epsilon_p v_{\max}^*) \tag{43}$$

$$= \frac{1 - \gamma^h}{1 - \gamma} (\epsilon_r + \gamma \epsilon_p v_{\max}^*). \tag{44}$$

Finally, by using (44) and recalling the definition of $\Delta_h$ (i.e., $\Delta_h \triangleq \max_{s,a} \Delta_h(s, a)$), we can directly prove that

$$\max_{s,a} \Delta_h(s, a) \triangleq \max_{s,a} |(\mathcal{T}^*)^h q^\pi(s, a) - (\mathcal{T}_{\hat{m}}^*)^h q^\pi(s, a)| \leq \frac{1 - \gamma^h}{1 - \gamma} (\epsilon_r + \gamma \epsilon_p v_{\max}^*). \tag{45}$$

$\square$

The result above bounds the difference between the action-value functions resulting from applying the Bellman optimality operator (defined with respect to the true model of the MDP) $h$ times, and the action-value function resulting from applying the Bellman optimality operator (defined with respect to an approximate model, $\hat{m}$) $h$ times. This allows us to precisely characterize—when repeatedly using the Bellman optimality operator over an initial action-value function—the maximum error incurred by using an approximate model of the environment.

The next two lemmas provide intermediate results that will be used in the proofs of Theorem 1 and Theorem 2.

**Lemma 3.** *Let $\Pi = \{\pi_i\}_{i=1}^n$ be a set of policies, $\{\hat{q}^{\pi_i}\}_{i=1}^n$ be approximations to their respective action-value functions, and $\hat{m} = (\hat{p}, \hat{r})$ be an approximate model such that, for all $\pi_i \in \Pi$ and for all $(s, a) \in \mathcal{S} \times \mathcal{A}$, the following holds:*

$$|q^{\pi_i}(s, a) - \hat{q}^{\pi_i}(s, a)| \leq \epsilon, \quad \sum_{s'} |p(s'|s, a) - \hat{p}(s'|s, a)| \leq \epsilon_p, \text{ and } |r(s, a) - \hat{r}(s, a)| \leq \epsilon_r. \tag{46}$$

*Recall the definition of $h$-GPI (Definition 3):*

$$\pi^{h\text{-}GPI}(s) \in \arg\max_{a \in \mathcal{A}} (\mathcal{T}_{\hat{m}}^*)^h \max_{\pi \in \Pi} \hat{q}^\pi(s, a). \tag{47}$$

*Then, it follows that:*

$$\mathcal{T}^{\pi^{h\text{-}GPI}} (\mathcal{T}_{\hat{m}}^*)^h \max_{\pi \in \Pi} \hat{q}^\pi(s, a) \geq (\mathcal{T}^*)^{h+1} \max_{\pi \in \Pi} q^\pi(s, a) - \gamma(\gamma^h \epsilon + c(\epsilon_r, \epsilon_p, h)). \tag{48}$$

*Proof.*

$$\mathcal{T}^{\pi^{h\text{-}GPI}} (\mathcal{T}_{\hat{m}}^*)^h \max_{\pi \in \Pi} \hat{q}^\pi(s, a) = \mathbb{E}\left[ r(s, a) + \gamma (\mathcal{T}_{\hat{m}}^*)^h \max_{\pi \in \Pi} \hat{q}^\pi(S_{t+1}, \pi^{h\text{-}GPI}(S_{t+1})) | S_t = s, A_t = a \right] \tag{49}$$

$$= \mathbb{E}\left[ r(s, a) + \gamma \max_{a' \in \mathcal{A}} (\mathcal{T}_{\hat{m}}^*)^h \max_{\pi \in \Pi} \hat{q}^\pi(S_{t+1}, a') | S_t = s, A_t = a \right] \tag{50}$$

$$\geq \mathbb{E}\left[ r(s, a) + \gamma \max_{a' \in \mathcal{A}} \left( (\mathcal{T}_{\hat{m}}^*)^h \max_{\pi \in \Pi} q^\pi(S_{t+1}, a') - \gamma^h \epsilon \right) | S_t = s, A_t = a \right] \tag{51}$$

$$\geq \mathbb{E}\left[ r(s, a) + \gamma \max_{a' \in \mathcal{A}} \left( (\mathcal{T}^*)^h \max_{\pi \in \Pi} q^\pi(S_{t+1}, a') - c(\epsilon_r, \epsilon_p, h) - \gamma^h \epsilon \right) | S_t = s, A_t = a \right] \tag{52}$$

$$\geq \mathbb{E}\left[ r(s, a) + \gamma \max_{a' \in \mathcal{A}} (\mathcal{T}^*)^h \max_{\pi \in \Pi} q^\pi(S_{t+1}, a') | S_t = s, A_t = a \right] - \gamma(\gamma^h \epsilon + c(\epsilon_r, \epsilon_p, h)) \tag{53}$$

$$= (\mathcal{T}^*)^{h+1} \max_{\pi \in \Pi} q^\pi(s, a) - \gamma(\gamma^h \epsilon + c(\epsilon_r, \epsilon_p, h)). \tag{54}$$

Above, (49) is due to the definition of the Bellman evaluation operator (21), and (50) is due to the definition of the $h$-GPI policy (47). Finally, (51) is due to Lemma 1 and (52) follows from the use of Lemma 2. □

**Lemma 4.** *Let $\Pi = \{\pi_i\}_{i=1}^n$ be a set of policies, $\{\hat{q}^{\pi_i}\}_{i=1}^n$ be approximations to their respective action-value functions, and $\hat{m} = (\hat{p}, \hat{r})$ be an approximate model such that, for all $\pi_i \in \Pi$, and for all $(s, a) \in \mathcal{S} \times \mathcal{A}$:*

$$|q^{\pi_i}(s,a) - \hat{q}^{\pi_i}(s,a)| \leq \epsilon, \quad \sum_{s'} |p(s'|s,a) - \hat{p}(s'|s,a)| \leq \epsilon_p, \text{ and } |r(s,a) - \hat{r}(s,a)| \leq \epsilon_r. \quad (55)$$

*Recall the definition of $h$-GPI (Definition 3):*

$$\pi^{h\text{-}GPI}(s) \in \arg\max_{a \in \mathcal{A}} (\mathcal{T}_{\hat{m}}^*)^h \max_{\pi \in \Pi} \hat{q}^\pi(s,a). \quad (56)$$

*Then,*

$$\mathcal{T}^{\pi^{h\text{-}GPI}}(\mathcal{T}_{\hat{m}}^*)^h \max_{\pi \in \Pi} \hat{q}^\pi(s,a) \geq (\mathcal{T}_{\hat{m}}^*)^h \max_{\pi \in \Pi} \hat{q}^\pi(s,a) - \gamma^h \epsilon(1+\gamma) - c(\epsilon_r, \epsilon_p, h)(1+\gamma). \quad (57)$$

*Proof.*

$$\mathcal{T}^{\pi^{h\text{-}GPI}}(\mathcal{T}_{\hat{m}}^*)^h \max_{\pi \in \Pi} \hat{q}^\pi(s,a) \geq (\mathcal{T}^*)^{h+1} \max_{\pi \in \Pi} q^\pi(s,a) - \gamma(\gamma^h \epsilon + c(\epsilon_r, \epsilon_p, h)) \quad (58)$$

$$\geq (\mathcal{T}^*)^h \max_{\pi \in \Pi} q^\pi(s,a) - \gamma(\gamma^h \epsilon + c(\epsilon_r, \epsilon_p, h)) \quad (59)$$

$$\geq (\mathcal{T}_{\hat{m}}^*)^h \max_{\pi \in \Pi} q^\pi(s,a) - c(\epsilon_r, \epsilon_p, h) - \gamma(\gamma^h \epsilon + c(\epsilon_r, \epsilon_p, h)) \quad (60)$$

$$\geq (\mathcal{T}_{\hat{m}}^*)^h \max_{\pi \in \Pi} \hat{q}^\pi(s,a) - \gamma^h \epsilon - c(\epsilon_r, \epsilon_p, h) - \gamma(\gamma^h \epsilon + c(\epsilon_r, \epsilon_p, h)) \quad (61)$$

$$= (\mathcal{T}_{\hat{m}}^*)^h \max_{\pi \in \Pi} \hat{q}^\pi(s,a) - \gamma^h \epsilon(1+\gamma) - c(\epsilon_r, \epsilon_p, h)(1+\gamma). \quad (62)$$

Above, (58) is by Lemma 3, (59) is due to the monotonicity of $\mathcal{T}^*$ (i.e. $\mathcal{T}^* q^\pi(s,a) \geq q^\pi(s,a)$), and (60) is by Lemma 2. □

We now use Lemmas 1, 2, and 4 to introduce Theorem 1, which extends the original GPI theorem (Barreto et al., 2017) and allows us to characterize the gains achievable by $h$-GPI as a function of approximation errors in both the agent's policy library, $\Pi$, and its (possibly learned) model, $\hat{m}$. Notice that Theorem 1 generalizes the original GPI theorem, which is recovered when $h = 0$.

**Theorem (1).** *Let $\Pi = \{\pi_i\}_{i=1}^n$ be a set of policies, $\{\hat{q}^{\pi_i}\}_{i=1}^n$ be approximations of their respective action-value functions, and $\hat{m} = (\hat{p}, \hat{r})$ be an approximate model such that (as previously-assumed in Lemmas 2 and 3), for all $\pi_i \in \Pi$ and all $(s, a) \in \mathcal{S} \times \mathcal{A}$,*

$$|q^{\pi_i}(s,a) - \hat{q}^{\pi_i}(s,a)| \leq \epsilon, \quad \sum_{s'} |p(s'|s,a) - \hat{p}(s'|s,a)| \leq \epsilon_p, \text{ and } |r(s,a) - \hat{r}(s,a)| \leq \epsilon_r. \quad (63)$$

*Recall once again the definition of $h$-GPI (Definition 3):*

$$\pi^{h\text{-}GPI}(s) \in \arg\max_{a \in \mathcal{A}} (\mathcal{T}_{\hat{m}}^*)^h \max_{\pi \in \Pi} \hat{q}^\pi(s,a). \quad (64)$$

*Then,*

$$q^{h\text{-}GPI}(s,a) \geq (\mathcal{T}^*)^h \max_{\pi \in \Pi} q^\pi(s,a) - \frac{2}{1-\gamma}(\gamma^h \epsilon + c(\epsilon_r, \epsilon_p, h)) \quad (65)$$

$$\geq \max_{\pi \in \Pi} q^\pi(s,a) - \frac{2}{1-\gamma}(\gamma^h \epsilon + c(\epsilon_r, \epsilon_p, h)), \quad (66)$$

*where $c(\epsilon_r, \epsilon_p, h) = \frac{1-\gamma^h}{1-\gamma}(\epsilon_r + \gamma\epsilon_p v_{max}^*)$.*

*Proof.*

$$q^{h\text{-GPI}}(s,a) = \lim_{k\to\infty}(\mathcal{T}^{\pi^{h\text{-GPI}}})^k(\mathcal{T}_{\hat{m}}^*)^h(\max_{\pi\in\Pi}\hat{q}^\pi(s,a)) \tag{67}$$

$$\geq (\mathcal{T}_{\hat{m}}^*)^h(\max_{\pi\in\Pi}\hat{q}^\pi(s,a)) - \lim_{k\to\infty}\sum_{i=0}^k \gamma^i(\gamma^h\epsilon(1+\gamma) + c(\epsilon_r,\epsilon_p,h)(1+\gamma)) \tag{68}$$

$$= (\mathcal{T}_{\hat{m}}^*)^h(\max_{\pi\in\Pi}\hat{q}^\pi(s,a)) - \frac{\gamma^h\epsilon(1+\gamma)}{1-\gamma} - \frac{c(\epsilon_r,\epsilon_p,h)(1+\gamma)}{1-\gamma} \tag{69}$$

$$\geq (\mathcal{T}_{\hat{m}}^*)^h(\max_{\pi\in\Pi}q^\pi(s,a)) - \gamma^h\epsilon - \frac{\gamma^h\epsilon(1+\gamma)}{1-\gamma} - \frac{c(\epsilon_r,\epsilon_p,h)(1+\gamma)}{1-\gamma} \tag{70}$$

$$\geq (\mathcal{T}^*)^h(\max_{\pi\in\Pi}q^\pi(s,a)) - c(\epsilon_r,\epsilon_p,h) - \gamma^h\epsilon - \frac{\gamma^h\epsilon(1+\gamma)}{1-\gamma} - \frac{c(\epsilon_r,\epsilon_p,h)(1+\gamma)}{1-\gamma} \tag{71}$$

$$= (\mathcal{T}^*)^h(\max_{\pi\in\Pi}q^\pi(s,a)) - \frac{2}{1-\gamma}(\gamma^h\epsilon + c(\epsilon_r,\epsilon_p,h)). \tag{72}$$

Above, (67) is due to the fixed-point of $\lim_{k\to\infty}(\mathcal{T}^{\pi^{h\text{-GPI}}})^k q(s,a)$ being $q^{h\text{-GPI}}(s,a)$ for any $q$. (68) is due to the repeated application of Lemma 4. Finally, (70) is due to Lemma 1, and (71) is due to Lemma 2. $\qquad\square$

> Theorem 1 characterizes the performance lower bound of $h$-GPI as a function of the number of planning steps, $h$, and the approximation errors in the agent's model and action-value functions (i.e., errors $\epsilon$, $\epsilon_p$, and $\epsilon_r$).

Note that since $(\mathcal{T}^*)^h \max_{\pi\in\Pi} q^\pi(s,a) \geq \max_{\pi\in\Pi} q^\pi(s,a)$, $h$-GPI's performance lower bound is strictly better than GPI's ($h = 0$) assuming no model approximation errors ($\epsilon_r = \epsilon_p = 0$). Furthermore, notice that as $h \to \infty$, $(\mathcal{T}^*)^h \max_{\pi\in\Pi} q^\pi(s,a)$ converges to $q^*$ and the approximation error term ($\gamma^h\epsilon$) in Equation (13) disappears. In other words, as $h$ increases, $h$-GPI's performance becomes arbitrarily less susceptible to sub-optimality in the agent's policy library. This implies that the planning horizon $h$ trades off between two conflicting objectives. On the one hand, increasing the planning horizon $h$ improves the performance of the $h$-GPI policy because the error term ($\epsilon$) associated with value function approximations errors becomes irrelevant. On the other hand, longer horizons worsen $h$-GPI's performance lower bound since they increase its dependency on errors arising from approximate models. Intuitively, for small values of $h$, the agent relies more heavily on the assumption that its estimates $\{\hat{q}^{\pi_i}\}_{i=1}^n$ are correct, given that these estimates are discounted by $\gamma^h$. For large values of $h$, by contrast, the approximate model error term, $c(\epsilon_r,\epsilon_p,h)$, increases, and the agent relies more heavily on the assumption that its (learned) model is approximately correct.

**Theorem (2).** *Let $\Pi = \{\pi_i^*\}_{i=1}^n$ be a set of optimal policies with respect to reward weights $\{\mathbf{w}_i\}_{i=1}^n$ and $\mathbf{w}$ be arbitrary reward weights. Let $\hat{m} = (\hat{p},\hat{r}_\mathbf{w})$ be an approximate model and $\{\hat{q}_\mathbf{w}^{\pi_i^*}\}_{i=1}^n$ be approximations to the action-value functions of policies in $\Pi$, under the reward function $r_\mathbf{w}$, such that for all $\pi_i \in \Pi$ and all $(s,a) \in \mathcal{S} \times \mathcal{A}$,*

$$|q_\mathbf{w}^{\pi_i^*}(s,a) - \hat{q}_\mathbf{w}^{\pi_i^*}(s,a)| \leq \epsilon, \quad \sum_{s'}|p(s'|s,a) - \hat{p}(s'|s,a)| \leq \epsilon_p, \text{ and } |r_\mathbf{w}(s,a) - \hat{r}_\mathbf{w}(s,a)| \leq \epsilon_r. \tag{73}$$

*We now extend the definition of $h$-GPI (Definition 3) to the case where this policy is defined under the assumption of an MPD with reward function $r_\mathbf{w}$:*

$$\pi^{h\text{-GPI}}(s;\mathbf{w}) \in \arg\max_{a\in\mathcal{A}}(\mathcal{T}_{\hat{m}}^*)^h \max_{\pi\in\Pi}\hat{q}_\mathbf{w}^\pi(s,a). \tag{74}$$

*Let $\phi_{max} \triangleq \max_{s,a}||\phi(s,a)||$. Then, it follows that*

$$q_\mathbf{w}^*(s,a) - q_\mathbf{w}^{h\text{-GPI}}(s,a) \leq \frac{2}{1-\gamma}(\phi_{max}\min_i||\mathbf{w}-\mathbf{w}_i|| + \gamma^h\epsilon + c(\epsilon_r,\epsilon_p,h)), \tag{75}$$

*where $c(\epsilon_r,\epsilon_p,h) = \frac{1-\gamma^h}{1-\gamma}(\epsilon_r + \gamma\epsilon_p v_{max}^*)$.*

*Proof.* The proof is an extension of Theorem 2 of Barreto et al. (2017) to the case where we use our Theorem 1 (i.e., we apply the bound introduced earlier to characterize the relation between the performance of $h$-GPI and the model and action-value function approximation errors), rather than using the GPI theorem, as defined by Barreto et al. (2017):

$$q_{\mathbf{w}}^*(s,a) - q_{\mathbf{w}}^{h\text{-GPI}}(s,a) \leq q_{\mathbf{w}}^*(s,a) - (\mathcal{T}^*)^h \max_{\pi \in \Pi} q_{\mathbf{w}}^\pi(s,a) + \frac{2}{1-\gamma}(\gamma^h \epsilon + c(\epsilon_r, \epsilon_p, h)) \tag{76}$$

$$\leq q_{\mathbf{w}}^*(s,a) - \max_{\pi \in \Pi} q_{\mathbf{w}}^\pi(s,a) + \frac{2}{1-\gamma}(\gamma^h \epsilon + c(\epsilon_r, \epsilon_p, h)) \tag{77}$$

$$\leq \frac{2}{1-\gamma} \min_i \max_{s,a} |r_{\mathbf{w}}(s,a) - r_{\mathbf{w}_i}(s,a)| + \frac{2}{1-\gamma}(\gamma^h \epsilon + c(\epsilon_r, \epsilon_p, h)) \tag{78}$$

$$= \frac{2}{1-\gamma} \min_i \max_{s,a} |\phi(s,a) \cdot \mathbf{w} - \phi(s,a) \cdot \mathbf{w}_i| + \frac{2}{1-\gamma}(\gamma^h \epsilon + c(\epsilon_r, \epsilon_p, h)) \tag{79}$$

$$= \frac{2}{1-\gamma} \min_i \max_{s,a} |\phi(s,a) \cdot (\mathbf{w} - \mathbf{w}_i)| + \frac{2}{1-\gamma}(\gamma^h \epsilon + c(\epsilon_r, \epsilon_p, h)) \tag{80}$$

$$\leq \frac{2}{1-\gamma} \min_i \max_{s,a} ||\phi(s,a)|| \, ||\mathbf{w} - \mathbf{w}_i|| + \frac{2}{1-\gamma}(\gamma^h \epsilon + c(\epsilon_r, \epsilon_p, h)) \tag{81}$$

$$= \frac{2}{1-\gamma} \phi_{\max} \min_i ||\mathbf{w} - \mathbf{w}_i|| + \frac{2}{1-\gamma}(\gamma^h \epsilon + c(\epsilon_r, \epsilon_p, h)) \tag{82}$$

$$= \frac{2}{1-\gamma}(\phi_{\max} \min_i ||\mathbf{w} - \mathbf{w}_i|| + \gamma^h \epsilon + c(\epsilon_r, \epsilon_p, h)). \tag{83}$$

Above, (76) is due to Theorem 1 and (77) is due to the monotonicity of $\mathcal{T}^*$. Finally, (78) is due to the application of Lemma 1 of Barreto et al. (2017), and (81) is due to the Cauchy-Schwarz's inequality. $\qquad\square$

Theorem 2, above, allows us to characterize $h$-GPI's performance (under approximation errors) with respect to the optimum action-value function, given an arbitrary reward function, $r_{\mathbf{w}}$:

> Theorem 2 precisely characterizes the optimality gap (i.e., the maximum difference) between the action-value function induced by $h$-GPI ($q_{\mathbf{w}}^{h\text{-GPI}}(s,a)$) and the optimal action-value function ($q_{\mathbf{w}}^*(s,a)$), as a function of *(i)* the reward weights $\{\mathbf{w}_i\}_{i=1}^n$ for which the policies in the agent's library, $\Pi$, are optimal; *(ii)* approximation errors in action-value functions $\{\hat{q}_{\mathbf{w}}^{\pi_i^*}\}_{i=1}^n$; and *(iii)* approximation errors in the model $\hat{m} = (\hat{p}, \hat{r}_{\mathbf{w}})$.

# B   Experimental details

In this section, we provide a detailed description of the practical implementation of $h$-GPI, as well as details of the domains used in our experiments. The code necessary to reproduce our results is available in the Supplemental Material.

## B.1   Computing $h$-GPI policies with SFs and Forward-Backward Dynamic Programming

In this section, we show how to extend the Forward-Backward Dynamic Programming (FB-DP) algorithm (Efroni et al., 2020) to the discounted-SFs setting, in order to compute $h$-GPI policies efficiently. The algorithm is described in Algorithm 3.

The algorithm works by first computing the sets of states achievable from the current state, $s$, after $t = 0, ..., h$ time steps (line 1 of Algorithm 3). The complete procedure is described in the Forward-Pass algorithm (Algorithm 2). Next, the algorithm for computing $h$-GPI policies (Algorithm 3) assigns the value of states reachable in $h$ using GPI bootstrapping with the SFs estimates (line 3). The values of states reachable in $t = 1, ..., h - 1$ time steps are then computed via dynamic programming (lines 4–5). Finally, the $h$-GPI action is identified in lines 6–8.

Regarding Algorithm 3, we first note that, given a state $s$, the algorithm's time complexity is $O(N|\mathcal{A}||\mathcal{S}_h^{\text{tot}}|)$, where $N$ is the maximal number of accessible states in one step (i.e., maximal "nearest neighbors" from any state), and $|\mathcal{S}_h^{\text{tot}}|$ is the total number of reachable states in $h$ time steps from state $s$.

Note that while we chose to extend the original FB-DP algorithm to design a practical procedure for computing $h$-GPI policies, due to its efficiency and closed-form formulation, other planning techniques could have been used (e.g., Monte Carlo Tree Search (Tesauro and Galperin, 1996; Silver et al., 2017)).

---

**Algorithm 2:** Forward-Pass

**Input :** Model $\hat{m} = (\hat{p}, \hat{\phi})$, planning horizon $h \geq 0$, state $s$.
1   $\mathcal{S}_0 \leftarrow \{s\}, \mathcal{S}_t \leftarrow \{\} \, \forall t \in \{1, \ldots, h\}$
2   **for** $t = 1, \ldots, h$ **do**
3     **for** $s_{t-1} \in \mathcal{S}_{t-1}$ **do**
4       $\mathcal{S}_t \leftarrow \mathcal{S}_t \cup$ $\{s' \mid \exists a \text{ s.t. } \hat{p}(s'|s_{t-1}, a) > 0\}$
5   **Return:** $\{\mathcal{S}_t\}_{t=0}^h$

---

**Algorithm 3:** $h$-GPI with SFs and FB-DP

**Input :** Model $\hat{m} = (\hat{p}, \hat{\phi})$, SFs $\{\hat{\psi}^{\pi_i}\}_{i=1}^n$, planning horizon $h \geq 0$, state $s$, reward vector $\mathbf{w}$.
1   $\{\mathcal{S}_t\}_{t=0}^h \leftarrow \text{ForwardPass}(\hat{m}, h, s)$
2   **for** $s \in \mathcal{S}_h$ **do**
3     $v_h(s) \leftarrow \max_{a \in \mathcal{A}} \max_{\pi \in \Pi} \hat{\psi}^{\pi}(s, a) \cdot \mathbf{w}$
4   **for** $t = h-1, \ldots, 1$ **do**
5     **for** $s \in \mathcal{S}_t$ **do**
6       $v_t(s) \leftarrow \max_{a \in \mathcal{A}}[\hat{\phi}(s, a) \cdot \mathbf{w} + \gamma \sum_{s'} \hat{p}(s'|s, a) v_{t+1}(s)]$
7   **for** $a \in \mathcal{A}$ **do**
8     $(\mathcal{T}_{\hat{m}}^*)^h \max_{\pi \in \Pi} \hat{q}_{\mathbf{w}}^{\pi}(s, a) \leftarrow$ $[\hat{\phi}(s, a) \cdot \mathbf{w} + \gamma \sum_{s'} \hat{p}(s'|s, a) v_1(s)]$
9   **Return:** $\arg\max_a (\mathcal{T}_{\hat{m}}^*)^h \max_{\pi \in \Pi} \hat{q}_{\mathbf{w}}^{\pi}(s, a)$

---

## B.2 Environments

Below, we describe the three domains used in our experiments—all of which are depicted in Figure 6.

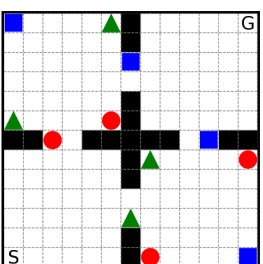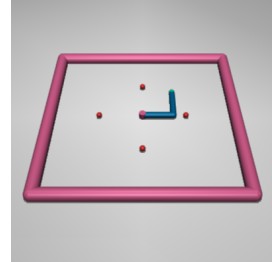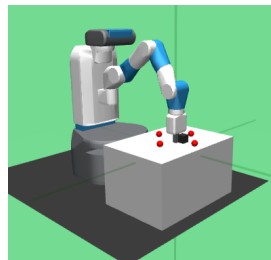

Figure 6: The FourRoom, Reacher, and FetchPush domains.

**FourRoom.** The Four Room domain (Barreto et al., 2017; Gimelfarb et al., 2021) is a gridworld with dimensions of $13 \times 13$, consisting of four rooms separated by walls. At each time step $t$, the agent occupies a cell and can move in one of four directions, denoted by the action set $\mathcal{A} = \{\text{up}, \text{down}, \text{left}, \text{right}\}$. Differently from related works, we introduced a 10% chance of the agent moving to a random direction, rather than the direction given by the selected action. This makes the state transition function, $p$, stochastic. If the destination cell happens to be a wall, the agent remains in its current cell. The grid contains three different types of objects, as illustrated in the leftmost figure in Figure 6. There are four instances of each object type distributed throughout the grid. The state space is defined as the concatenation of the agent's current x-y coordinates and a set of binary variables indicating whether each object has been picked up or not: $\mathcal{S} = \{0, 1, \ldots, 12\}^2 \times \{0, 1\}^{12}$. The features $\phi(s, a, s') \in \{0, 1\}^3$ are one-hot encoded vectors that indicate the type of object present in the current cell. If there are no objects in the current cell, the corresponding features are set to zero. The goal cell which the agent has to reach is located at the upper-right of the map. It contains one instance of each object, and entering this cell results in the termination of the episode. In this domain,

a discount factor of $\gamma = 0.95$ was used. In our experiments, we used the implementation available on MO-Gymnasium (Alegre et al., 2022b).

**Reacher.** The Reacher domain is a classic domain in the SFs literature (Barreto et al., 2017; Gimelfarb et al., 2021; Nemecek and Parr, 2021; Alegre et al., 2022a). It consists of a two-joint robot arm that must reach different target locations with the tip of its arm (see the middle figure in Figure 6). The agent's state space $\mathcal{S} \subset \mathbb{R}^6$ consists of the sine and cosine of the angles of the central and elbow joints, as well as their angular velocities. The agent's initial state is the one shown in the middle figure in Figure 6. The action space, originally continuous, is discretized using 3 bins per dimension corresponding to maximum positive torque (+1), negative torque (-1), and zero torque for each actuator. This results in a total of 9 possible actions: $\mathcal{A} = \{-1, 0, +1\}^2$. Each feature $\phi(s, a, s') \in \mathbb{R}^4$ is defined as $\phi_i(s, a, s') = 1 - 4\Delta(\text{target}_i), i = 1...4$, where $\Delta(\text{target}_i)$ is the Euclidean distance between the tip of the robot's arm and the $i$-th target's location. We used a discount factor of $\gamma = 0.9$ in this domain. We used the implementation of this domain available on MO-Gymnasium (Alegre et al., 2022b), which is based on the MuJoco robotics simulator (Todorov et al., 2012).

**FetchPush.** Finally, we extended the FetchPush domain (Plappert et al., 2018), which consists of a Fetch robotic arm that must move a block to a given target position on top of a table by pushing the block with its gripper (see the rightmost figure in Figure 6). Importantly, the state space of this domain, $\mathcal{S} \subset \mathbb{R}^{19}$, is high-dimensional, and its dynamics are significantly more complex than that of Reacher's. In our experiments, we removed from the states all information related to the position of targets, as these are irrelevant since targets have fixed locations. We discretized the action space similarly to how this was done when defining the Reacher domain. The reward features $\phi(s, a, s') \in \mathbb{R}^4$ correspond to the negative distances between the block and each of the four target locations on the table. Our implementation of this domain is an adaptation of the one available in Gymnasium-Robotics (de Lazcano et al., 2023).

### B.3 Parameters

In the FourRoom experiments, the SFs of each policy were learned similarly as in Alegre et al. (2022a), using Q-learning and 5 Dyna updates per time step. Each policy was trained for $10^6$ time steps using a learning rate of $0.1$ and epsilon-greedy exploration with a probability of selecting a random action linearly decayed from 1 to 0.05 during half of the training period.

In the Reacher and FetchPush domains, we used multi-layer perception (MLP) neural networks to learn universal successor feature approximators (USFAs) (Borsa et al., 2019). The USFAs, $\hat{\psi}(s, a, \mathbf{w})$, were modeled using MLPs with 4 layers of 256 neurons and ReLU non-linear activations. We used the Adam optimizer (Kingma and Ba, 2015) with a learning rate of $3 \cdot 10^{-4}$, and mini-batches of size 256. Recall that (as discussed in the main paper), we used vectors that form the standard basis of $\mathbb{R}^d$ as training reward weights, $\mathcal{M}$, where $d$ is the dimension of the reward features vector $\phi$. In our implementation, we also adopted popular DQN extensions to speed up and stabilize learning, such as prioritized experience replay (Schaul et al., 2016; Fujimoto et al., 2020) and Double Q-learning (Hasselt et al., 2016) with a target neural network updated after every 200 time steps. In both domains (Reacher and FetchPush), we trained the USFAs for a total of $2 \cdot 10^5$ time steps. In Reacher, the epsilon-greedy exploration rate was kept fixed at 0.05, while in FetchPush, it was linearly decayed from 1 to 0.05.

We used an ensemble of $n=7$ probabilistic neural networks in both the Reacher and FetchPush environments to learn the model $m_\varphi(S_{t+1}, \phi_t | S_t, A_t)$. In the Reacher domain, each network was an MLP with 3 layers with 200 neurons each. In the FetchPush domain, we used an MLP with 4 layers with 400 neurons each. The probabilistic neural networks used to approximate the model were trained with early stopping based on a holdout validation subset with instances drawn from the experience buffer $\mathcal{B}$—as commonly done when training such networks (Chua et al., 2018; Janner et al., 2019). In Figure 7, we depict the mean validation loss of the learned models both in the Reacher and FetchPush domains. As can be seen, the learned model rapidly becomes fairly accurate.

**Compute.** For the tabular experiments, we used an Intel i7-8700 CPU @ 3.20GHz computer with 32GB of RAM. For the experiments involving the function approximation setting, we used computers with NVIDIA A100-PCIE-40GB GPUs. For each random seed, the training period used for training

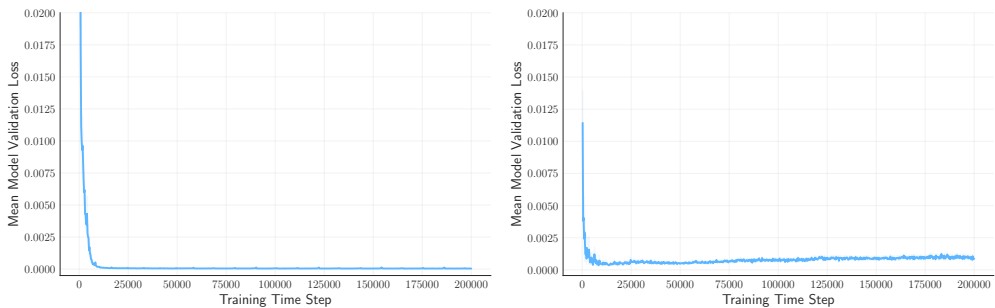

Figure 7: Mean validation loss of the learned environment model, $\hat{m}_\varphi(s', \phi|s, a)$, throughout the training phase during which $\Pi$ is constructed (Left: Reacher domain; Right: FetchPush domain).

USFAs was of approximately 6.5 hours. We used the JAX library (Bradbury et al., 2018) for the implementation of neural networks.

## C    Related work

In this section, we discuss in more detail works that are related to $h$-GPI.

**GPI and SFs.** Previous works have extended GPI to safe RL (Gimelfarb et al., 2021; Feng et al., 2023), maximum-entropy RL (Hunt et al., 2019), unsupervised RL (Hansen et al., 2020), and hierarchical RL (e.g., via the options framework) (Barreto et al., 2019; Machado et al., 2023). Recently, Thakoor et al. (2022) introduced Geometric GPI (GGPI). GGPI uses geometric horizon models (GHMs) to learn the discounted future state-visitation distribution induced by particular policies to rapidly evaluate policies under a given, known reward function. The authors show that performing GPI over a particular type of non-stationary policy produces behaviors that outperform those in the agent's library policy. $h$-GPI, by contrast, learns a different type of model: an environment model, which is used to perform planning—i.e., action selection—rather than policy evaluation. Additionally, GGPI requires learning separate GHMs for each policy in the library, whereas $h$-GPI can operate with a single model that predicts the next reward features, alongside a single USFA. Secondly, GGPI assumes that the reward function is known *a priori*, while we exploit SFs to generalize over all linear rewards given reward features. Bagot et al. (2023) introduced GPI-Tree Search (GPI-TS), which is closely related to $h$-GPI. GPI-TS uses GPI bootstrapping as backup value estimates at the leaf nodes of a Monte Carlo tree search. However, GPI-TS does not employ SFs and was only employed in single-task settings. Moreover, it assumes an oracle model of the environment. We, by contrast, employ learned models to perform zero-shot transfer over multiple tasks, and we show how approximation errors in the model affect the performance of the $h$-GPI policy (see Theorems 1 and 2). Kim et al. (2022) introduced Constrained GPI (CGPI), which uses lower and upper bounds of the expected return of the optimal policy for a new task to constrain the action-values used when selecting the GPI policy's actions. Although CGPI is robust with respect to generalization errors in the values predicted by USFAs for new target tasks, it is sensitive to approximation errors in the values of the training tasks. $h$-GPI, by contrast, can inherently deal with approximation errors in the values of source/base policies. Other works have studied methods for constructing a policy library for use with GPI (Zahavy et al., 2021; Nemecek and Parr, 2021; Alver and Precup, 2022; Alegre et al., 2022a). These methods solve important orthogonal problems, and could potentially be combined with $h$-GPI.

**Model-based RL and SFs.** Russek et al. (2017) proposed a Dyna-style algorithm that performs background planning to update tabular successor representations (SRs) (Dayan, 1993). Momennejad et al. (2017) argued that this algorithm could better model human behavior on tasks with changing rewards when compared to pure model-free or model-based approaches. In this paper, by contrast, we employed online planning and leveraged GPI and SFs—both of which generalize SRs to continuous-state problems. Other works have studied methods for learning SF-based representations (Lehnert and Littman, 2020; Touati and Ollivier, 2021). Studying the behavior of $h$-GPI under different types of reward feature representations is an interesting direction for future research. We believe that our work

also shares similarities with the investigation performed by Tomov et al. (2021) on how model-free and model-based mechanisms are combined in human behavior.

**Multi-step RL algorithms.** Multi-step RL methods were extensively studied in the context of policy evaluation, both in the model-free (Hessel et al., 2018; van Hasselt et al., 2018) and the model-based settings (Yao et al., 2009; Janner et al., 2019). Model value expansion algorithms (Feinberg et al., 2018; Buckman et al., 2018; Abbas et al., 2020) are a representative example of the latter type of approach. In this paper, by contrast, we introduced a multi-step method for policy improvement, designed specifically to tackle transfer-learning settings. GX-Chen et al. (2022) introduced the $\eta$-return mixture, a new backup target for better credit assignment, which combines bootstrapping with standard value estimates and SFs, as a function of a parameter $\eta$. Efroni et al. (2018) studied multi-step greedy versions of the classic policy iteration and value iteration (Bertsekas and Tsitsiklis, 1996) dynamic programming (DP) algorithms. Efroni et al. (2020) proposed a multi-step greedy real-time DP algorithm that replaces 1-step greedy policies (used in policy improvement) with an $h$-step lookahead policy. The technique introduced in our paper is also related to the works of Sikchi et al. (2022) and Hansen et al. (2022), which combine planning and bootstrapping with a learned value estimate. However, unlike $h$-GPI, the abovementioned approaches did not address the multi-policy and zero-shot transfer settings. Another interesting use of multi-step prediction models is the work of Filos et al. (2022). In that work, the authors defined a measure of epistemic uncertainty based on the disagreement between the values resulting from applying the model-induced Bellman evaluation operator for different numbers of steps. Finally, we also emphasize to the interested reader that there exists neuroscientific evidence that multi-step planning occurs in the brains of humans and other animals (Miller and Venditto, 2021).

