# OpenReview forum: "Multi-Step Generalized Policy Improvement by Leveraging Approximate Models"
_NeurIPS.cc/2023/Conference — NeurIPS 2023 poster_

### Official Review · Reviewer_obSw · 2023-07-03

**Soundness:** 3 good
**Presentation:** 4 excellent
**Contribution:** 4 excellent
**Rating:** 7
**Confidence:** 4

**Summary:**

The authors present an to the Generalized Policy Improvement (GPI) framework, where the agent is able to leverage approximate models to derive a policy in a zero-shot fashion that adapts to unseen tasks related to others it has previously solved.
The paper derives an h-step planning horizon style action selection where the agent is able to trade-off long-term planning horizons for potentially compounding approximation errors in its update.

The authors provide theoretical results on the bound of improvement on their proposed method as it relates to model approximation error, and show that their approach is a consistent improvement over GPI in several standard RL benchmarks.

**Strengths:**

- The paper extends the GPI framework to leverage approximate models of the problem at hand. This provides a natural extension of the model-free approach of GPI, to a model based approach that allows an agent to take advantage of long-horizon planning when selecting actions.

- The authors provide a theoretical bound on the improvement of their approach as it relates to the model approximation error.

- The authors also presented a thorough experimental section with several standard benchmarks in the tabular and continuous state cases. These results show a consistent improvement over standard GPI.

**Weaknesses:**

- The main weakness I see is not of the method itself, but perhaps on how to apply it in practice. The theoretical results bounds the improvement of GPI based on the model approximation error, and in the limit one could set the h-step to 0 and recover GPI.
However, since one of the main benefits is the potential of zero-shot adaptation to novel tasks, it is difficult to see how one would estimate the model error in practice when trying to solve a new task.
How would you know what h would be right when solving a new problem? According to Fig 3, the right h value seems to be quite important.

**Questions:**

- See weaknesses

- Line 197-198, it states the "increasing" the planning horizon improves performance because the error term becomes irrelevant. Did you mean to say decreasing? Why would increasing horizon make error terms irrelevant? Wouldn't the error compound and therefore become more relevant to the performance?


Nitpick:

- Notation in Eq. 7, and subsequent uses of max_{a' \in A}(T).....max_{a \in A} is not a function applied to the operator T itself, but rather to the result of the operator applied to q.
This should probably be max_{a' \in A}( T*^h q(St+1, a;))....notice the parenthesis.


- Eq 10, why the sudden change from pi to mu to refer to a policy? Is there any special meaning to it? If so, please clarify, if not, please pick a notation a be consistent with it.

- line 116 reads a bit awkward, I think there are some typos there.
- line 151 that's not the correct use of e.g.

**Limitations:**

- The authors that in practicality it would also be needed a way to assess the discrepancy between the model approximation and the true model to determine what the value of h should be.

---

> ### Author Rebuttal · Authors · 2023-08-09
>
> We thank the reviewer for the positive assessment of our work, and for highlighting the theoretical and experimental contributions of the paper.
>
> ### How to identify the best possible value for $h$ ?
> This is a great question relating to our formal bounds and the information needed to deploy our method. The formal bounds we introduced serve to mathematically quantify, e.g., how errors in the model and in the agent's value function approximators impact its performance, as a function of the planning horizon $h$. The goal of proving these bounds is to mathematically guarantee (for instance) that as $h$ increases, our method becomes arbitrarily less susceptible to sub-optimality in the agent's prior knowledge for solving previously-experienced tasks.
>
> Formally proving such bounds allows us to provide formal guarantees about the behavior and optimality properties of $h$-GPI as a function of the amount of planning resources that are given to the agent, thereby demonstrating *(i)* that ours is not a heuristic method for zero-shot transfer; and that *(ii)* its performance degrades gracefully as a function of imperfections in the agent's model and approximators.
>
> Below we comment on the process of selecting effective values for $h$.
>
> * The reviewer is correct that if one wished to estimate the *optimal* value of $h$, they would need access to the true value of all error terms. In practice (as is the case with many other machine learning techniques) *deploying* our method does not require knowledge about such quantities. Instead, $h$ is seen as an adjustable hyperparameter of $h$-GPI.
> * Since $h$ regulates how the algorithm interpolates between purely model-free approaches (like GPI) and fully model-based RL (like Dynamic Programming), it we it as similar in nature to the $\lambda$ parameter of TD($\lambda)$. This latter hyper-parameter also regulates the amount of interpolation—in this case, between purely model-free Monte Carlo approaches and fully model-based Dynamic Programming techniques.
> * In practice, even though formal bounds relating approximation errors, performance, and $h$ (in our case) or bias, variance, and $\lambda$ (in the case of TD($\lambda$), both algorithms are deployed with the understanding that, in practice, their hyper-parameters $h$ and $\lambda$ are domain-dependent hyper-parameters. These are in practice adjusted empirically or given domain knowledge, rather than computed based on the formal equations describing the formal properties of each method.
> * Having said that, the reviewer is absolutely right that it is important to determine if, in practice, our method's performance is not overly sensitive to $h$. In our experimental section, we show that $h$-GPI outperforms competing methods essentially for *all* the values of $h$ that we tested (concretely, in all experiments and comparisons we performed, competing methods outperformed $h$-GPI in only 2\% of the cases even when considering possibly adversarial values of $h$). This shows that the algorithm is robust to the choice of its (only) hyperparameter. Extending the method to adaptively identify the best value of $h$ is an interesting direction for future work, and we will discuss it in our Future Work section.
>
> ### On the impact of model errors and planning horizon on $h$-GPI's performance
> * In the sentence mentioned by the reviewer (lines 197--198), we are referring to errors, $\epsilon$, in the agent's approximations of successor features. In this case, the longer the horizon, the more the agent relies on the model and the smaller the effect of possible successor feature approximation errors when performing GPI after $h$ steps. In particular, any performance estimation errors due to imperfect SFs will be discounted by $\gamma^h$, thereby having a vanishing impact on the agent's performance as $h$ increases.
> * The reviewer is correct that increasing the horizon $h$ makes model errors compound—$h$ trade-offs between these two possible sources of error (model errors or errors in the agent's approximators). The impact of increasing $h$ in the agent's performance, as a function of increasingly imperfect models, is characterized in Theorem 1 and lines 189--191.
> * Regarding the latter point (compounding modeling errors), this effect can be observed in the FetchPush experiments (Figure B). Here, $h$-GPI's mean performance decreases slightly due to model errors for some intermediate values of $h$. However, further increasing $h$ past those values causes $h$-GPI to once again outperform competitors since, intuitively, the benefits of being able to perform transfer and generalization in future steps outweigh the downsides of short-term model predictions.
>
> ### Feedback on presentation
> * We thank the reviewer for their attention to detail and for the feedback regarding a few minor suggestions to improve presentation and clarity. We will address all of them in the updated version of the paper.

---

> > ### Comment · Reviewer_obSw · 2023-08-15
> > **Response to Authors**
> >
> > I thank the authors for their thorough responses.
> > After going over the rebuttal and other reviews, I am maintaining my original score.

---

### Official Review · Reviewer_CAn3 · 2023-07-05

**Soundness:** 3 good
**Presentation:** 3 good
**Contribution:** 2 fair
**Rating:** 6
**Confidence:** 3

**Summary:**

The paper presents a hybrid model-free and model-based policy improvement method for zero-shot policy generalization with successor features. The generalized value function is calculated using a generalized version of the multi-step Bellman operator, where a learned model is used in the rollout to predict future states and reward features.

**Strengths:**

The idea to use multistep updates for generalized policy evaluation and improvement is novel to my knowledge. The paper is well written and relatively easy to understand. The authors carefully report detailed statistics for their experiments. I think that generally, the approach is a good idea and can be impactful if supported by the right set of experiments.

**Weaknesses:**

I am not sure about the meaning of the experimental results in section 4. As I understand it, the h-GPI approach amounts to planning ahead for h steps, and then obtaining the "rest" of the estimated future value for times after h using model-free successor features. If the learned model is relatively accurate, and the successor features are, compared to that, less accurate, then naturally the results improve as h increases and the decision-making relies more on the model over the successor features (as the authors also state after theorem 1). Likewise, it is unsurprising that simply ignoring the remaining value after h steps (as the SF-MPC baseline does iiuc, see footnote on page 8) performs worse than h-GPI. I think that a fair baseline would be SF-MPC for horizons much longer than h (the authors state themselves that SF-MPC underperforms since it "requires long-term reasoning to collect items which are far from the agent").

Alternatively, one could make the argument that the "cut-off" done by h-GPI after h steps is a performance advantage over SF-MPC with much longer horizons, since rollouts can be shorter. However, I don't think that the authors are currently trying to make that argument; runtimes are barely mentioned (except for a brief discussion in the limitations section, but only in comparison to vanilla GPI).

I would therefore suggest comparing to SF-MPC with much longer horizons, and if this should outperform h-GPI, maybe consider arguing from the perspective of run-time performance, supported by additional experiments on that.

Minor issues: Should the reference in line 311 really be to figure 4, or should it be to figure 3? Also, it would probably help to quickly mention in the caption of figure 3 in what environment these experiments were run.

**Questions:**

- (See also above): What would happen if you increased the horizon for SF-MPC further?
- How good is the dynamics model in the experiments for Figs 2,3, 4, and 5? Maybe it would be beneficial to report prediction errors (although not absolutely necessary if results for SF-MPC for long horizons are shown)
- If the model quality is not perfect, is there a sweet-spot h at which the performance of h-GPI starts to decrease again with increasing h?


**Limitations:**

Limitations are discussed relatively briefly, it might be worth broadening this section a little.

---

> ### Author Rebuttal · Authors · 2023-08-10
>
> We thank the reviewer for the constructive feedback and for the suggestion of new experiments. Below we discuss how our newly added experiments clarify your raised concern.
>
> ### Comparing $h$-GPI with SF-MPC with longer horizons
> The reviewer asked about the meaning of the experimental results presented in Section 4. As described in lines 263--264, the objective of the experiments is "to evaluate the effectiveness of $h$-GPI as a method for zero-shot policy transfer".
>
> The reviewer, subsequently, argued that $h$-GPI might outperform SF-MPC primarily because they think SF-MPC may have been allowed to plan for an insufficient number of timesteps. **This is not the case** for the following reasons:
> 1. In our experiments, we *always* allow SF-MPC to plan for a number of steps that is higher than or equal to the number of steps that we allow $h$-GPI to plan (see the footnote on page 8). In other words, *SF-MPC is always given strictly more planning resources than $h$-GPI*, in all experiments. Therefore, even when SF-MPC is allowed to reason for significantly longer horizons than $h$-GPI, $h$-GPI still consistently outperforms it.
> 2. To make this point clearer, we performed additional experiments and added novel results showing that even if we allow SF-MPC to plan for *twice as many steps* as $h$-GPI, $h$-GPI still outperforms it. Please see Figures A and B in the attached one-page PDF.
> 3. As an example, in Figure A (FourRoom) one can see that even with a small planning horizon of $h=2$, $h$-GPI is capable of matching the performance of SF-MPC even when SF-MPC is allowed to plan for **ten times** longer (ie., $h=20$).
> 4. Similarly, the experiment depicted in Figure B (FetchPush) compares both methods when SF-MPC is allowed to plan for up to $h=15$ steps—that is, **three times** longer than the maximum horizon we allow $h$-GPI to plan. Again, $h$-GPI consistently outperforms it (and other baselines) even when allowed to plan for significantly fewer steps.
>
> We hope these results better clarify and emphasize the advantages of $h$-GPI over fully model-based planning methods as well as over various state-of-the-art GPI baselines.
> ### How accurate is the learned model in your experiments?
> To provide the reviewer with concrete data on this, we have included a new figure (Figure C) in the additional one-page PDF. This figure depicts the mean validation loss of the learned models both in the Reacher and FetchPush domains. As can be seen, the learned model rapidly becomes fairly accurate. Importantly, therefore, even with a well-trained model, SF-MPC (*even when given more than twice the amount of planning resources than $h$-GPI*) cannot outperform the technique we introduced in this paper.
> ### Impact of the value of $h$ and model errors on $h$-GPI's performance
> Thank you for bringing up this question. The reviewer is correct that when models are imperfect, $h$-GPI's performance may decrease under long horizons (large values of $h$). This effect can be observed in the FetchPush experiments (Figure 5). Here, $h$-GPI's mean performance decreases slightly due to model errors for some intermediate values of $h$. Importantly, however, further increasing $h$ past those values causes $h$-GPI to once again outperform competitors since, intuitively, the benefits of being able to perform transfer and generalization in future steps outweigh the downsides of short-term model predictions.
> ### Feedback on presentation
> We thank the reviewer for their attention to detail and for the feedback regarding two minor suggestions to improve presentation and clarity. The first point brought up by the reviewer was, indeed, a minor typo; regarding the second suggestion given by the reviewer, we will update the figure's caption to clarify it relates to the FourRoom domain. We will address both of these suggestions in the updated version of the paper.
> ### Are "unsurprising" experimental results a weakness of the paper?
> The reviewer argues that, in retrospect, combining planning steps and the generalization and transfer power of successor features is bound to result in a powerful method that performs well (they say that some experimental results are "unsurprising"). They argue that this is a weakness of the paper. We strongly disagree that experimental results being (from a given reviewer's perspective) "unsurprising" is a *weakness* of a paper.
> - The goal of our experiments was to provide strong empirical supporting the formal properties of $h$-GPI—which they did.
> - We have presented strong additional results addressing the reviewer's primary concern regarding the reasons why $h$-GPI outperformed SF-MPC.
> - Besides the strong empirical evidence of our method's performance, our contribution lies in that we introduced a method that is novel, principled, with various non-trivial formal performance guarantees, and that outperforms many strong, state-of-the-art competitors.
> - Up to this point, the problem of designing a principled method capable of combining imperfect models and the generalization power of GPI had remained an open problem in the field.
> ### Final comments
> We would like to once again thank the reviewer for their constructive feedback and clarification questions. The main—and only—concern brought up by the reviewer was related to whether $h$-GPI outperforms other methods due to unfair advantages related to planning horizons. We performed additional experiments to tackle this concern and provided further and strong support to our claim that $h$-GPI outperforms all competitors.
> * We trust the strong results discussed above fully address the reviewer's sole concern.
> * *Considering this resolution, and also in light of the positive feedback expressed, e.g., by obSw and MMCf, we would be thankful if the reviewer could revisit their evaluation of this paper.*
> * *We hope that our detailed response and the strengths identified may lead you to align your evaluation with an Acceptance decision.*

---

> > ### Comment · Reviewer_CAn3 · 2023-08-16
> > **My main concern has been adressed**
> >
> > I thank the authors for their thorough rebuttal. I believe that my main concern (potentially unfairly low horizon for SF-MPC) has been properly addressed by their clarifications and additional experiments. My other concerns were all derivative of this. I will therefore raise my score.

---

> ### Comment · Reviewer_CAn3 · 2023-08-14
> **I accidentally pasted my summary into the "Strengths" section, and now corrected this in my review (I will consider and answer the rebuttal later)**
>
> I just realized I accidentally copied over the "summary" section into the "strength" section when submitting my comments. I edited my review to correct this.
>
> My apologies to the authors for that.
>
> I will also read the author's rebuttal later this week.

---

### Official Review · Reviewer_Y3y3 · 2023-07-07

**Soundness:** 3 good
**Presentation:** 3 good
**Contribution:** 2 fair
**Rating:** 5
**Confidence:** 4

**Summary:**

This paper studies the combination of model-based planning/rl and model free generalized policy improvement and proposes h-GPI. The authors provide both theoretical and experimental results to support the method.

**Strengths:**

1. This paper is clearly written and easy to follow.
2. The motivation to study the intersection between model-based planning and GPI is also straightforward and clear.
3. The theoretical results are thorough, and are correct in my perspective.

**Weaknesses:**

1. For readers who are not familiar with GPI, Definition 1 is a bit confusing. What is $\pi'$ used for? In equation 5, $\pi'$ does not appear. Does that mean $\pi^{GPI}$ satisfies equation 4?
2. Can the authors explain how the policy set $\Pi$ is obtained?
3. Can the results generalize to the algorithm that replaces GPI to standard policy iteration/TD? It seems that h-GPI is a generalization of h-step model value expansion [1]. If so, I am interested in the comparison of h-GPI to MVE in both theory and experiments. Otherwise, the novelty compared to previous model-based value expansion works can be limited.
4. The current theoretical results only cover the bias of the proposed algorithm, which meets our expectation as model-free and model-based combinations require the trade-off of h. It would be more comprehensive if more analysis on the sample/computational complexity can be added.

[1] Vladimir Feinberg et al. Model-Based Value Estimation for Efficient Model-Free Reinforcement Learning


**Questions:**

See the weakness above.

**Limitations:**

See the weakness above.

---

> ### Author Rebuttal · Authors · 2023-08-09
>
> We thank the reviewer for acknowledging that the paper is clearly written, that the theoretical results are thorough and correct, and that both theoretical and experimental results support the method. Below, we carefully answer all concerns raised in your review:
>
> ### On the definition of GPI
> Thank you for the feedback on our discussion about GPI. We will include extra discussion on the intuition underlying this definition. This will be done by reminding the reader that GPI is the process by which an agent can combine policies in a given set, $\Pi$, to obtain a new policy, $\pi'$, such that its performance is at least as good as the performance of any policy in the original set, $\Pi$ (lines 116-117). This property is formalized in Eq. 4. One way to identify a policy $\pi'$ satisfying such property is via the GPI procedure, which computes a policy $\pi^\text{GPI}$ according to Eq. 5. This is described in lines 123-124.
> ### How can $\Pi$ be constructed?
> This is a great question, and one of the open problems in the field. In theory, any arbitrary set of policies, $\Pi$, may be used when performing GPI. In the Related Work, we discuss four recently-proposed methods introducing different heuristics for constructing $\Pi$ (lines 342--344). In our experiments, as discussed in lines 299-300, we constructed $\Pi$ via a commonly-used approach described in Borsa et al. (2019) and Kim et al. (2022). In the FourRoom domain, this approach identifies the base policies in $\Pi$ by solving tasks whose reward functions are defined by weight vectors $\mathbf{w} \in \\{ [1,0,0], [0,1,0], [0,0,1] \\}$. We will emphasize that $h$-GPI is agnostic to the set $\Pi$ used.
>
> ### Can GPI be replaced with standard TD techniques?
> Our method uses GPI to directly determine the performance of a policy when optimizing a given reward function. Determining the performance of a policy can be done via any standard policy evaluation technique. We use GPI due to its strong generalization and transfer properties, which allow us to *directly* determine a novel policy's performance—unlike TD methods, which demand the agent to interact with the environment and would thus be more sample inefficient.
> ### Sample/computational complexity
> * Regarding the computational complexity of $h$-GPI, it primarily depends on the underlying planning technique used. In this paper, e.g., we extended the FB-DP algorithm (Efroni et al., 2020), allowing $h$-lookahead policies to be computed in real-time (see pseudocode in Appendix B.1.). Concretely, given an initial state $s$, the time complexity of this algorithm is $O(N|A||S_{h}^{\text{tot}}|)$, where $N$ is the maximal number of accessible states in one step, and $S_{h}^{\text{tot}}$ is the number of reachable states in $h$ time steps from state $s$. It is worth noting that the complexity of performing the other significant step of $h$-GPI—GPI bootstrapping—is negligible compared to planning, as it only requires querying the learned $q$-value policies in the library $\Pi$.
> * Regarding the sample complexity of *deploying* $h$-GPI, we recall that our paper addresses a zero-shot transfer setting. In this context, agents *cannot* collect any further samples from the environment when tackling a new task. Instead, they must rely solely on previously-acquired information—regardless of how it was obtained—to identify potential solutions. Since no samples can be collected at deployment time, defining or quantifying sample complexity becomes non-trivial. A more meaningful way to characterize how the method scales might be to analyze how quickly it reduces errors in its solutions as a function of the properties of the knowledge available to the agent (e.g., value function approx. errors). This is explored in Section 3. We appreciate you bringing up this point, and we believe that adding a corresponding discussion will enhance the clarity of the paper.
>
> ### Can $h$-GPI be compared with MVE?
> This is a good question. Although $h$-GPI and MVE are, on a high level, based on a similar motivation (i.e., exploiting possibly imperfect models of the environment), they solve qualitatively different problems.
> - As discussed in the Related Work, MVE algorithms (Feinberg et al., 2018) are model-based algorithms for *policy evaluation* in the single-task setting. We, by contrast, introduce a model-based algorithm for *policy improvement* in the multi-task setting. Since they are solving qualitatively different problems, and under different assumptions, a comparison is not possible.
> - We will, in our Future Work section, discuss potential ways in which $h$-GPI may be used to extend MVE's capabilities. In particular, it may be used to extend MVE so that MVE becomes applicable not only in single-task settings, but also in multi-task problems.
> Furthermore, $h$-GPI might potentially be employed to incorporate a novel policy improvement procedure to MVE so it can learn (rather than merely evaluate) effective policies.
>
> ### Final comments
> We sincerely thank the reviewer for their constructive insights! In your review:
> * You asked for clarification regarding two minor points.
> * You pointed out a relevant high-level similarity between $h$-GPI and MVE and proposed a direct comparison. We hope our discussion on how these methods solve distinct problems addresses your question and clarify why an empirical comparison, in the setting we address in this paper, would not be possible even in principle.
>
> We trust that our responses have addressed your concerns and provided the necessary clarification on the questions you had and on your primary concern.
> - *Based on this additional information, and in line with the positive strengths you noted (regarding our setting, motivation, and theoretical results), we kindly request that you reconsider your evaluation.*
> - *We would be glad to address any other questions you may have. Otherwise, we hope that this enhanced understanding of our work may align with an Acceptance decision for this paper.*

---

> > ### Comment · Reviewer_Y3y3 · 2023-08-11
> > **Response to authors**
> >
> > I thank the authors for the response. I decide to raise my score to 5.

---

### Official Review · Reviewer_MMCf · 2023-07-25

**Soundness:** 3 good
**Presentation:** 2 fair
**Contribution:** 3 good
**Rating:** 6
**Confidence:** 2

**Summary:**

This paper proposes a new hybrid solution of model-free and fully model-based reinforcement learning solutions for planning. It introduces a multi-step extension of generalized policy improvement by having certain planning steps to approximate model behavior, increasing the efficiency of the model. It is followed by zero-shot policy transfer.

**Strengths:**

1. Although the originality of the problem itself is not new, it does address the problem of working efficiency of current methods.
2. The algorithm itself is very well-written, starting from basics of RL, MDPs and extended to the contribution itself.

**Weaknesses:**

1. The method itself lacks subjective comparison with other fully model-based or model-free solutions under same learning environment.
2. Although the method addresses the problem of efficiency, it is not fully motivated and justified (why it should be useful and advancing from earlier model-free/model-based methods and why this is necessary?)
3. The method is applied to two main setups - statistical one and deep learning based one. However, the setup looks a bit hand-crafted and lacks earlier reference. I do not think they comes from no-where - authors may address earlier works and add some general descriptions of those two settings, before getting into details.

**Questions:**

1. Why h-GPI should be useful and advancing from earlier model-free/model-based methods and why this is necessary?
2. From high-level perspective, the proposed method looks like a fine-tuning of an approximators. So do you think this can be applied to "conventional" deep model-based planning methods? If so, do you think hGPI can still outperform?
3. Do you think the method itself will be heavily rely on the generalization power of the universal SF feature approximator, rather than the model itself? It gives me this kind of instinct when reading the experimental sections.

**Limitations:**

The authors listed some limitations of their methods at the very last section. Potential negative societal impact may not be applicable.

---

> ### Author Rebuttal · Authors · 2023-08-10
>
> We thank the reviewer for the positive assessment of our work. Below, we address their remaining questions:
> ### On $h$-GPI's underlying motivations and advantages
> Thank you for your feedback that further intuitive discussion on the motivation and advantages of $h$-GPI may be provided. The goal of $h$-GPI is to perform zero-shot transfer in RL—in a principled way—by exploiting possibly imperfect models of the environment. $h$-GPI advances upon existing model-free and model-based methods, in the zero-shot transfer setting, in various ways.
> * Existing *model-free* zero-shot transfer methods often rely on techniques rooted in GPI and SFs. These techniques analyze a library of policies—each solving a particular task—and identify which action the agent should take. Although computationally efficient, they cannot exploit models of the environment that the agent may have access to, thus rendering them less sample efficient.
> * Fully *model-based* algorithms can—if given perfect knowledge about the environment—identify optimal policies. However, they are typically computationally intensive, and the assumption that exact models are available is seldom true. In reality, these techniques may fail catastrophically since modeling errors are known to compound with the number of actions taken by the agent.
> * $h$-GPI simultaneously improves upon these extremes. It implements a multi-step extension of GPI that interpolates between these extremes and exploits the best of both worlds. The interpolation is controlled by $h$, which regulates the amount of time the agent has to reason.
> * As $h$ increases, $h$-GPI exploits the agent's model and becomes arbitrarily less susceptible to sub-optimality in the agent's prior knowledge for solving previously-experienced tasks—see Theorem 1.
> * Finally, *(i)* $h$-GPI is more computationally efficient than fully model-based approaches, since it does not have to construct complete, possibly long-horizon plans; *(ii)* it is less susceptible to model errors, since it avoids compounding modeling errors by exploiting GPI to directly predict the agent's performance in future states; and *(iii)* it is more efficient than purely model-free methods, since it identifies promising actions by looking ahead into the future.
>
> The discussion about the points above is currently presented in the Introduction and Section 3. We will follow the reviewer's suggestion and further emphasize these points.
> ### Information about domains used in experiments
> We thank the reviewer for this suggestion. We agree that this change will benefit readers who may be unfamiliar with the field.
> * We will emphasize more strongly that a complete and detailed description of our experimental setup—both for the tabular/statistical case, and for the deep RL case—is already available in Appendix B2.
> * We will update the paper to include further references, as requested by the reviewer, to emphasize that all settings investigated in this paper are consolidated in the literature of SFs\&GPI. Appendix B2 discusses this and provides a complete and thorough set of references for each experiment. The FourRoom and Reacher domains have been investigated by many other works (Barreto et al. (2017); Gimelfarb et al. (2021); Nemecek and Parr (2021); Alegre et al. (2022)). Both of them are well-known experimental setups used to evaluate GPI-based algorithms both in the tabular setting (discrete state spaces) and in the deep RL setting (continuous states and function approximation with NNs).
> * We will move parts of the description of the domains from Appendix B2 to the main body, for the benefit of readers unfamiliar with particular details.
> * We will clarify that multi-task version of the FetchPush domain was introduced by us to showcase the capabilities of our method in a setting with more complex state dynamics.
> ### Combining $h$-GPI with "conventional" deep model-based planning algorithms
> * The reviewer is correct that $h$-GPI can be instantiated in settings that combine various types of model-based RL algorithms—including those using deep model-based techniques. In our experiments with the Reacher and FetchPush domains, we do combine planning steps (inherent to the way $h$-GPI works) and deep learning algorithms—necessary as these domains have continuous states.
> * Regarding the possibility of combining $h$-GPI with other deep model-based algorithms, it suffices to compute the first term of Eq. 10 (responsible for the online planning step) using any other techniques of interest.
> * There are many benefits to doing so compared to deploying these methods directly. If the agent only has access to imperfect environment models, $h$-GPI empowers the agent to identify the best possible action by first planning with the approximate model for $h$ steps, and then exploiting the transfer and generalization properties of GPI to *directly* estimate the return from future states. This capability is something that standard deep model-based planning algorithms currently do not possess.
> * We appreciate the reviewer's insight and will update the paper to discuss this promising strategy for extending existing algorithms with $h$-GPI.
> ### Does performance rely primarily on the generalization power of USFAs?
> * $h$-GPI's performance does not primarily rely on universal successor feature approximators (USFAs). This observation is supported by our empirical results. In the FourRoom domain, for example, $h$-GPI significantly improves both upon standard GPI and MPC (for numerous values of $h$) *even though USFAs are not used*. In the other domains, we allow *both* $h$-GPI and GPI to use USFAs. In this case, even though all methods can exploit the generalization power of USFAs, $h$-GPI still outperforms them. Therefore, the benefits of $h$-GPI are not primarily due to USFAs; by contrast, they arise from its strong formal guarantees regarding the performances that are achievable when interpolating between GPI and model-based approaches.

---

> > ### Comment · Reviewer_MMCf · 2023-08-20
> >
> > Thanks to the reviewer for the comments. I think I did not put as much emphasis as to main content when going through the appendix. However, as you suggested, it might be good to migrate some of the stuff from Appendix into the main content.
> >
> > Regarding the possibility of combining -GPI with other deep model-based algorithms, it suffices to compute the first term of Eq. 10 (responsible for the online planning step) using any other techniques of interest.
> > ------I think this is my only remaining problem. I would not ask for further experiments, but I need further motivation on constraining the scale of experiment to current level.

---

> > > ### Author Response · Authors · 2023-08-21
> > >
> > > We are glad to hear that our previous response has addressed the reviewer's questions and concerns.
> > >
> > > The reviewer's last question asks for further discussion on our previous comment that:
> > >
> > > > *"(...) regarding the possibility of combining $h$-GPI with other deep model-based algorithms, it suffices to compute the first term of Eq. 10 (responsible for the online planning step) using any other techniques of interest"*.
> > >
> > > We thank the reviewer for bringing this point up. Please find our comments on it below.
> > >
> > > 1) To tackle the tabular setting, we extended the state-of-the-art FB-DP algorithm (Efroni et al., 2020). This technique allows lookahead policies to be computed in real-time (please see the pseudocode in Appendix B). In particular, we exploited such an approach because it enables us to design and deploy an *exact* method, with well-characterized computational complexity, and known empirical efficiency. We are not aware of other methods with similar properties or guarantees that could have been used in our setting.
> > > 2) In the function approximation setting, we computed the first term of Eq. 10 using a planning technique similar to that proposed by Chua et al. (2018). This method, based on an ensemble of neural networks, aligns with current state-of-the-art deep planning techniques. Hafner et al. (2019), for example, proposed and used a similar MPC planning approach—though one involving a more complex neural network architecture tailored for image-based tasks.
> > > 3) Our experiments aim to demonstrate that $h$-GPI consistently outperforms several baselines and competitors—even when they are given more planning/computational resources. Importantly, we emphasize that, to ensure fairness, all experiments were designed so that both our approach ($h$-GPI) and all competing SF-MPC baselines used the *same underlying model and planning technique*. For this reason, the particular choice of planning method does not affect our conclusions.
> > >
> > > We once again thank the reviewer for raising this point. We believe that a discussion of alternative approaches to planning and the reasons underlying our algorithmic choices will benefit the reader. We will further extend the current discussion in Section 3.1 to incorporate the reviewer's suggestions.

---

### Author Rebuttal · Authors · 2023-08-10

We would like to thank the reviewers for their positive feedback and for the constructive suggestions for improving the paper.

In summary, one of the main concerns raised by reviewers was the need to compare GPI with SF-MPC under longer horizons. To address this concern, we have performed additional experiments to support our claims and the properties of our method. The corresponding results are shown in the additional attached one-page PDF file. Importantly:
1. Our novel results show that even if we allow SF-MPC to plan for *twice as many steps* as $h$-GPI, $h$-GPI still outperforms it. Please see Figures A and B in the attached one-page PDF.
3. As an example, in Figure A (FourRoom) one can see that even with a small planning horizon of $h=2$, $h$-GPI is capable of matching the performance of SF-MPC even when SF-MPC is allowed to plan for **ten times** longer (ie., $h=20$).
4. Similarly, the experiment depicted in Figure B (FetchPush) compares both methods when SF-MPC is allowed to plan for up to $h=15$ steps—that is, **three times** longer than the maximum horizon we allow $h$-GPI to plan. Again, $h$-GPI consistently outperforms it (and other baselines) even when allowed to plan for significantly fewer steps.

In addition, we have carefully answered *all* other questions regarding our method properties—as well as minor clarifications regarding related works—in the answers provided as rebuttal to each of the reviewers.

We hope our detailed comments, as well as the additional strong empirical results discussed above, fully address the reviewers' main concerns.

---

### Author Response · Authors · 2023-08-19
**Post Rebuttal — Thank You**

We thank the reviewers for the encouraging comments about our work, for highlighting the impact of our theoretical contributions, and for acknowledging that our experimental results (including new, additional experiments requested by reviewers) empirically support our method’s theoretical guarantees and show that it consistently outperforms competitors.

We are also happy to hear that reviewers said that our responses successfully addressed their questions and concerns. We sincerely appreciate them revisiting their scores to reflect their view that our paper’s contributions are consistent with an Acceptance vote.

---

### Decision · Program_Chairs · 2023-09-21

**Decision:**

Accept (poster)

**Comment:**

The paper considers improving accuracy compared to generalized policy improvement in zero-shot transfer learning setting, as a function of the horizon. The contribution seems to be both interesting and significant, both from a theoretical perspective as well as in terms of applications. There are few-limitations, however, such as how to properly determine the most effective horizon length to use.